# Actin and an unconventional myosin motor, TgMyoF, control the organization and dynamics of the endomembrane network in *Toxoplasma gondii*

**Romain Carmeille**, **Porfirio Schiano Lomoriello**, **Parvathi M. Devarakonda**, **Jacob A. Kellermeier**, **Aoife T. Heaslip** *

Department of Cell and Molecular Biology, University of Connecticut, Storrs, Connecticut, United States of America

* aoife.heaslip@uconn.edu

**Data Availability Statement:** All relevant data are within the manuscript and its Supporting Information files.

## Abstract

*Toxoplasma gondii* is an obligate intracellular parasite that relies on three distinct secretory organelles, the micronemes, rhoptries, and dense granules, for parasite survival and disease pathogenesis. Secretory proteins destined for these organelles are synthesized in the endoplasmic reticulum (ER) and sequentially trafficked through a highly polarized endomembrane network that consists of the Golgi and multiple post-Golgi compartments. Currently, little is known about how the parasite cytoskeleton controls the positioning of the organelles in this pathway, or how vesicular cargo is trafficked between organelles. Here we show that F-actin and an unconventional myosin motor, TgMyoF, control the dynamics and organization of the organelles in the secretory pathway, specifically ER tubule movement, apical positioning of the Golgi and post-Golgi compartments, apical positioning of the rhoptries, and finally, the directed transport of Rab6-positive and Rop1-positive vesicles. Thus, this study identifies TgMyoF and actin as the key cytoskeletal components that organize the endomembrane system in *T. gondii*.

## Author summary

Endomembrane trafficking is a vital cellular process in all eukaryotic cells. In most cases the molecular motors myosin, kinesin, and dynein transport cargo including vesicles, organelles and transcripts along actin and microtubule filaments in a manner analogous to a train moving on its tracks. For the unicellular eukaryote *Toxoplasma gondii*, the accurate trafficking of proteins through the endomembrane system is vital for parasite survival and pathogenicity. However, the mechanisms of cargo transport in this parasite are poorly understood. In this study, we fluorescently labeled multiple endomembrane organelles and imaged their movements using live cell microscopy. We demonstrate that filamentous actin and an unconventional myosin motor named TgMyoF control both the positioning of organelles in this pathway and the movement of transport vesicles throughout the

**Funding:** This work was funded by National Institutes of allergy and infectious disease 1R21AI121885-01 awarded to ATH, National institute of general medical sciences, 1R35GM138316-01 awarded to ATH, and the University of Connecticut Research Excellence Program awarded to ATH. The funders had no role in study design, data collection and interpretation, the decision to submit the work for publication or manuscript preparation. The authors declare that no competing interests exist.

**Competing interests:** The authors have declared that no competing interests exist.

parasite cytosol. This data provides new insight into the mechanisms of cargo transport in this important pathogen and expands our understanding of the biological roles of actin in the intracellular phase of the parasite's growth cycle.

## Introduction

*Toxoplasma gondii* is a member of the phylum Apicomplexa, which contains over 5000 species of parasites that cause substantial morbidity and mortality worldwide [1,2]. *T. gondii* can cause life-threatening disease in immunocompromised individuals and when infection occurs in utero [3–5]. Additionally, *T. gondii* is estimated to cause persistent life-long infection in 10–70% of the world's population depending on geographic location [6].

*T. gondii* is an obligate intracellular parasite, and thus parasite survival and disease pathogenesis rely on the lytic cycle of the parasite. The lytic cycle involves host cell invasion, parasite replication within a specialized vacuole termed the parasitophorous vacuole (PV), and host cell egress that results in the destruction of the infected cells (reviewed by [7]). To complete this lytic cycle, the parasite relies on three specialized secretory organelles, the micronemes, rhoptries, and dense granules. Micronemes are small vesicles that are localized predominately at the parasite's apical end [8] and are important for parasite motility, attachment and initiating invasion (Reviewed by [9]). Rhoptries are larger club shaped organelles that contain two sub-sets of proteins (rhoptry bulb proteins (ROPs) and rhoptry neck proteins (RONs)) categorized based on their functions and location within the rhoptry [10]. After initial attachment to the host cell, RONs contribute to the formation of the moving junction, a ring structure that aids in the propulsion of the parasite into the host cell [11–15]. Once invasion is initiated ROPs and dense granule proteins (GRAs) are secreted into the host cell where they control the organization and structure of the PV, and modulate host gene expression and immune response pathways [10,16–18].

Secretory proteins destined for these distinct organelles are synthesized in the endoplasmic reticulum (ER) and must sequentially traverse multiple intermediate compartments within *T. gondii's* highly polarized endomembrane system before ultimately arriving at their final destination (Fig 1A). Newly synthesized proteins are first trafficked to the Golgi which is located adjacent to the nucleus at the apical end of the parasite. Dense granules are formed from post-Golgi vesicles [19,20] while proteins destined for the micronemes and rhoptries are trafficked through one or more post-Golgi compartments (PGCs), although the exact route taken by each secretory protein has not been elucidated and the function of each PGC has not been fully defined. Two of these compartments are marked by Rab5a and Rab7 and are referred to as the endosome-like compartments, as these proteins are markers of the early and late endosomes in higher eukaryotes [21]. While the function of the Rab7 compartment is not known [22], overexpression of Rab5a results in the mislocalization of rhoptry proteins and a subset of microneme proteins, to the dense granules, indicating a role in protein sorting [23]. Another Rab GTPase, Rab6, is thought to localize to the Golgi and dense granules although the function of Rab6 has not been defined [24]. Syntaxin 6 (TgSyn6) marks yet another distinct PGC and appears to have a role in retrograde trafficking between the Rab5 and Rab7 compartments and the trans-Golgi network [25]. A fifth compartment, the plant-like vacuole (VAC) has similarities to the central vacuole in plants. This compartment contains lytic enzymes that are important for proteolytic processing of the micronemes, ion regulation and processing endocytic material [26–28]. Endocytosis has only recently been described in *T. gondii* [29,30]. After uptake, material is trafficked to the

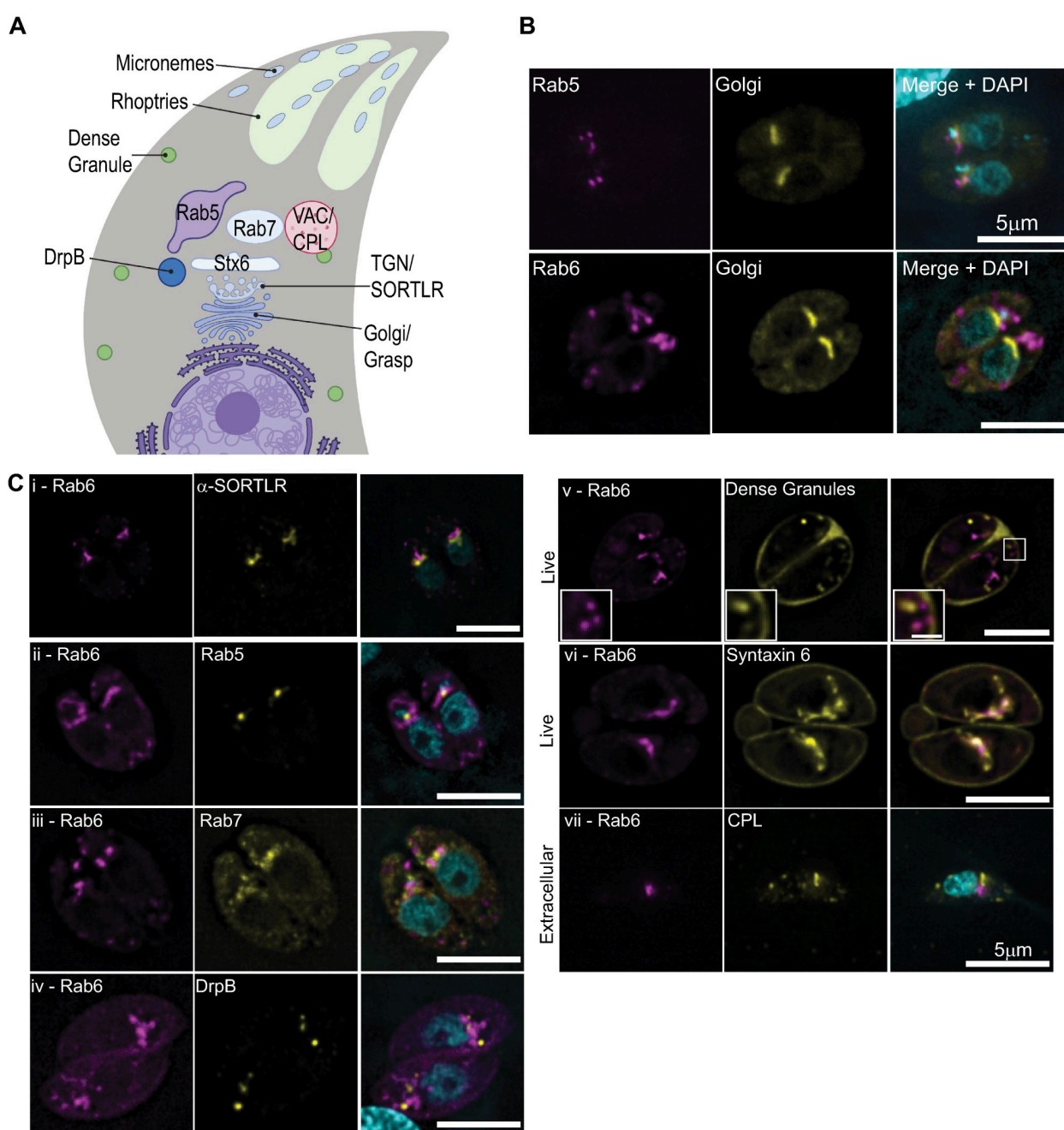

**Fig 1. Rab6 and syntaxin6 co-localize in a post-Golgi compartment.** (A) Schematic diagram of the endomembrane system in *T. gondii*. Created with BioRender.com. (B) RH parasites expressing Grasp55-mCherry (Golgi marker; yellow) and NeonFP-Rab5 (magenta; upper panel) or EmGFP-Rab6 (magenta; lower panel) were fixed and stained with DAPI (cyan). (C) RH parasites expressing EmGFP-Rab6 (magenta; panel i) or AppleFP-Rab6 (magenta; panels ii-vii) and markers of the endomembrane system as indicated (yellow). Nuclei were stained with DAPI (cyan). Panels i-iv are deconvolved images of a single focal plane from fixed intracellular parasites. Panels v-vi are a single focal plane from live intracellular parasites expressing AppleFP-Rab6 and SAG1ΔGPI-GFP (marker for the dense granules) or Syn6-GFP respectively. Panel vi is a single extracellular parasite expressing AppleFP-Rab6 that fixed and stained with an anti-CPL antibody and DAPI. Scale bar = 5μm. Inset scale bar in panel v is 1 μm.

VAC [31,32]. Although the function of the PGCs have not been fully defined, these compartments mark a point of intersection between the biosynthetic and endocytic pathways and are a hub for protein trafficking in the parasite.

The location of the PGCs adjacent to the apically positioned Golgi is presumed to optimize the transport of newly synthesized proteins to the parasite's apical end [22]. However, the cytoskeletal factors which control endomembrane positioning have not been identified. In mammalian cells, vesicle transport and organelle positioning are controlled by the microtubule cytoskeleton and associated kinesin and dynein motors [33,34]. In contrast, budding yeast uses bundled F-actin filaments as tracks for myosin V-based transport of vesicles, organelles and localizing RNA transcripts [35]. *T. gondii* contains 22 highly-stable microtubules (MTs), that are strictly localized at the parasite pellicle [36–38] (Fig 1A) and do not associate with organelles in the parasite cytosol. Moreover, microtubule depolymerization with oryzalin had no effect on dense granule transport [39] and thus, are likely not involved in organelle positioning or vesicle transport between the post-Golgi compartments.

*T. gondii* has a single actin gene (TgAct1) that has 83% similarity with mammalian β and γ actin isoforms [40]. The organization of the actin cytoskeleton in *T. gondii* remained elusive for many years, most likely because of the low propensity of TgAct1 to bind phalloidin, the gold standard reagent used to image actin in other cell types [41,42]. This inability to visualize F-actin *in vivo*, in addition to biochemical studies which suggested that TgAct1 was incapable of forming long stable filaments *in vitro* led to the idea that short TgAct1 filaments formed only transiently during parasite motility and host cell invasion (reviewed by [43]). However, more recent studies have uncovered new biological roles for TgAct1 including inheritance of the apicoplast (a non-photosynthetic plastid organelle) [44], dense granule transport [39] and microneme recycling during parasite division [45]. Additionally, an unconventional myosin motor, TgMyoF, was shown to be required for apicoplast inheritance and dense granule transport [39,44]. With the development of a new reagent, the Actin chromobody (ActinCB) [46], actin filaments have now been visualized in both the parasite cytosol [46–48] and in a tubular network that connects individual parasites [46,49]. These *in vivo* observations are consistent with recent *in vitro* experiments by Lu and colleagues who used a total internal reflectance (TIRF) microscopy-based approach to demonstrate that Pfact1, actin from the closely related apicomplexan parasite *Plasmodium falciparum*, is capable of transiently forming long filaments up to 30μm in length [50]. Collectively, these findings have significantly altered our understandings of both the functions and organization of Apicomplexan actin.

This study aimed to define the role of actin and TgMyoF in regulating vesicle trafficking and organelle positioning within the endomembrane pathway. Our data demonstrates that both of these proteins are required for the apical positioning and morphology of the Golgi and post-Golgi compartments, ER tubule movement, and transport of Rab6-positive and Rop1-positive vesicles. These results indicate that this acto-myosin system is vital for controlling the organization of the endomembrane system in *T. gondii* and uncovers new biological roles of actin in the intracellular phase of the parasite's growth cycle.

## Results

### Rab6 colocalizes with syntaxin6 in a post-Golgi compartment

To investigate how the morphology and dynamics of the PGCs is controlled by the *T. gondii* cytoskeleton, we expressed NeonGreenFP-Rab5a (referred to subsequently as Neon-Rab5a) and EmeraldFP-Rab6 (EmGFP-Rab6) along with Grasp55-mCherryFP, a marker of the cis-Golgi [23,24,51,52]. As expected, Neon-Rab5 did not co-localize with Grasp55-mCherryFP and was found in a distinct compartment adjacent to the Golgi [23] (Fig 1B; upper panel). Surprisingly, Rab6 also did not co-localize with the cis-Golgi as previously reported [24] but rather localized to a compartment apical to the cis-Golgi (Fig 1B; lower panel). Since Rab6 was previously reported to localize to the Golgi, we transfected RH strain parasites with an

EmGFP-Rab6 expression construct and after 18 hours of intracellular growth, fixed and stained using an anti-SORTLR antibody, a marker of the trans-Golgi [53]. EmGFP-Rab6 and SORTLR did not co-localize, and although the two proteins are very closely aligned, SORTLR was consistently closer to the nucleus than EmGFP-Rab6 indicating that Rab6 is found in a post-Golgi compartment (Fig 1C-panel i).

To determine if Rab6 colocalizes with markers of the other post-Golgi compartments, we transfected parasites with an AppleFP-Rab6 expression construct along with Neon-Rab5, Neon-Rab7, Syn6-GFP, DrpB-GFP and SAG1ΔGPI-GFP, a marker for the dense granules [23,31,54,55] and grew parasites for ~18 hours before fixation or live cell imaging (Fig 1C; panels ii-vi). Additionally, extracellular parasites expressing EmGFP-Rab6 were fixed and stained with an anti-TgCPL antibody, a marker of the VAC (Fig 1C; panel vii) [27]. Rab6 and Syn6 localize within the same post-Golgi compartment. Syn6 was also localized to the parasite periphery but no peripheral staining of Rab6 was observed (Fig 1C-panel vi) [25]. No colocalization was observed between Rab6 and the other proteins analyzed (Fig 1C) including the dense granules which were previously reported to contain Rab6 on their surface [25] (Fig 1C; panel v).

Live cell imaging of parasites expressing Syn6-GFP and AppleFP-Rab6 demonstrates that these proteins occupy distinct sub-domains within the same tubular-vesicular compartment (Fig 2A and 2B and S1 Video). Line scan analysis indicates that Syn6 and Rab6 are both found in the "vesicular" domain of the compartment (Fig 2B; magenta arrow) while Rab6 positive/ Syn6 negative (Rab6(+)/Syn6(-)) tubules extend from this "vesicular" domain (Fig 2B; white bracket). Rab6(+) vesicles can be seen budding from the tip of the tubular extensions (Fig 2C). Rab6(+)/Syn6(-) vesicles are distributed throughout the parasite cytosol (Fig 2B arrowhead and S1 video) and exhibited directed, presumably motor-driven, motion.

## Rab6 vesicle transport is dependent on F-actin

To further characterize the dynamics and morphology of the Rab6 compartment, we imaged parasites expressing EmGFP-Rab6 using live cell microscopy with a temporal resolution of 100ms (Fig 3A and S2 Video). The Rab6 compartment is dynamic and undergoes constant rearrangement as indicated by images taken after 5, 10, and 15 seconds of imaging (Fig 3B and S2 Video) and we frequently observed Rab6 vesicles budding from this compartment (S1A and S1B Fig). Rab6 vesicles exhibited directed movement throughout the cytosol as illustrated by time lapse images and kymographs (Fig 3C). As we observed previously for dense granules, vesicle movement is bidirectional and vesicles move towards both the apical and basal ends of the parasite; in fact, individual vesicles frequently change their direction of movement (Figs 3C and S1). This motion was reminiscent of the directed actin-based motion exhibited by dense granules [39]. Using the Fiji-plugin MTrackJ we tracked the movement of vesicles exhibiting directed movement, which was defined as vesicles moving in the same direction for at least 10 frames [49,56] and we quantified the number of times we observe vesicles moving in a directed manner (directed runs) per parasite per minute (referred to subsequently as run frequency). Tracking the movement of Rab6 vesicles revealed velocities and run-lengths of 0.92±0.01μm/s and 1.6±0.03μm respectively (S1 Table).

Given the similarities between Rab6 vesicle movement and actin dependent dense granule movement [39], we sought to determine if actin was required for Rab6 vesicle transport. We treated intracellular parasites expressing EmGFP-Rab6 with cytochalasin D (CD) for 60 minutes to depolymerize F-actin before imaging (Fig 3D and S2 Video). We observed two phenotypes associated with the loss of F-actin: First, instead of a single Rab6 compartment at the parasites apical end, we observe multiple large Rab6 structures throughout the parasite cytosol,

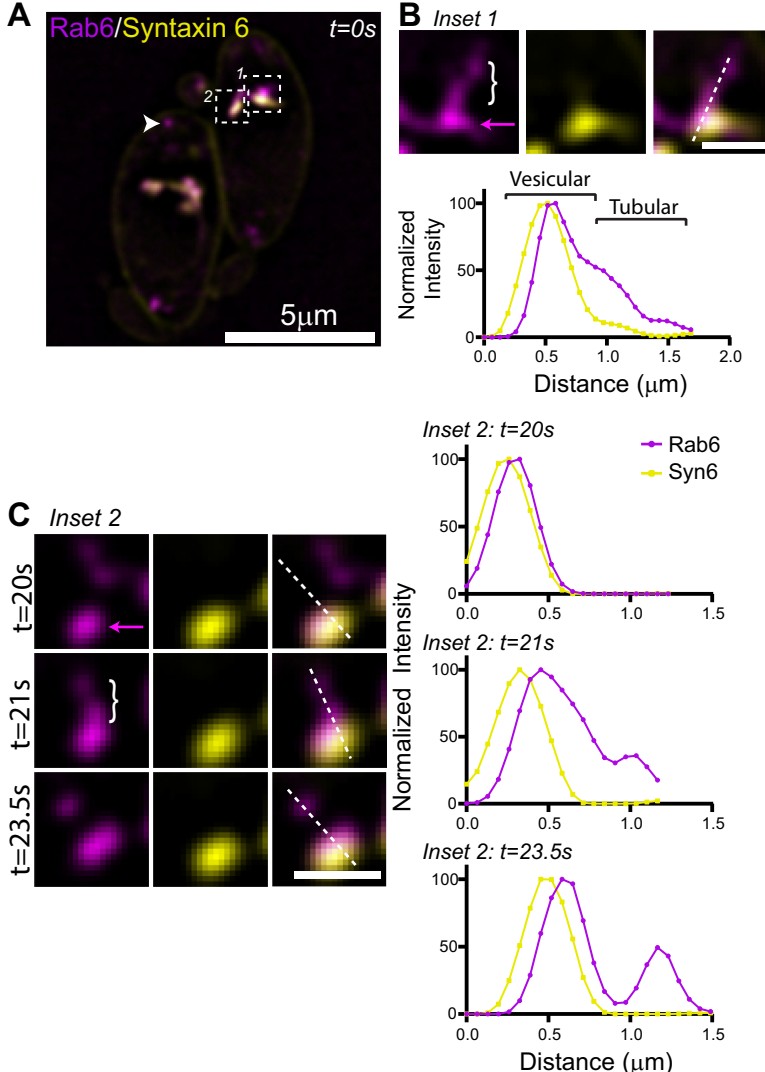

**Fig 2. The Rab6 and syntaxin6 PGC is dynamic.** (A) Live imaging of RH parasites expressing AppleFP-Rab6 and Syn6-GFP. Rab6 vesicles indicated with the arrow head. Areas in the white box were used to make inset images in panels (B) and (C). Scale bar = 5µm. (B) *Upper panel*. Inset 1 from panel (A). Dashed line was used to make line scan in lower panel. Bracket indicates Rab6 positive and Syn6 negative tubule. Magenta arrow indicates vesicular compartment containing both Rab6 and Syn6. *Lower panel*. Line scan of Rab6 and Syn6. Shift in peaks between Rab6 and Syn6 likely due to sequential image acquisition between the two imaging channels. Scale bar is 1µm. (C) *Left panel*. Image from Inset 2 from panel (A), taken at times indicated. Bracket indicates Rab6 positive and Syn6 negative tubule. Magenta arrow indicates vesicular compartment containing both Rab6 and Syn6. Dashed lines were used to make line scan in right panel. *Right panel*. Line scans of Rab6 and Syn6 at times indicated. Scale bar is 1µm.

including the basal end, suggesting that in the absence of F-actin the Rab6 compartment becomes fragmented (Fig 3D). Second, the dynamic tubular morphology of the EmGFP-Rab6 compartment is lost and the compartment remained static throughout the 60-second imaging period (Fig 3E). Vesicle formation from the Rab6 compartment was perturbed in CD treated parasites (Fig 3D–3K). The number of Rab6 vesicles in the parasite cytosol decreases from an average of 8±0.5 in control parasites (RH parasites treated with DMSO) to 3±0.3 after CD treatment (Fig 3K). Similarly, the number of directed runs exhibited by Rab6 vesicles decreased from 6±0.45 in control to less than 1 after CD treatment (Fig 3F–3L).

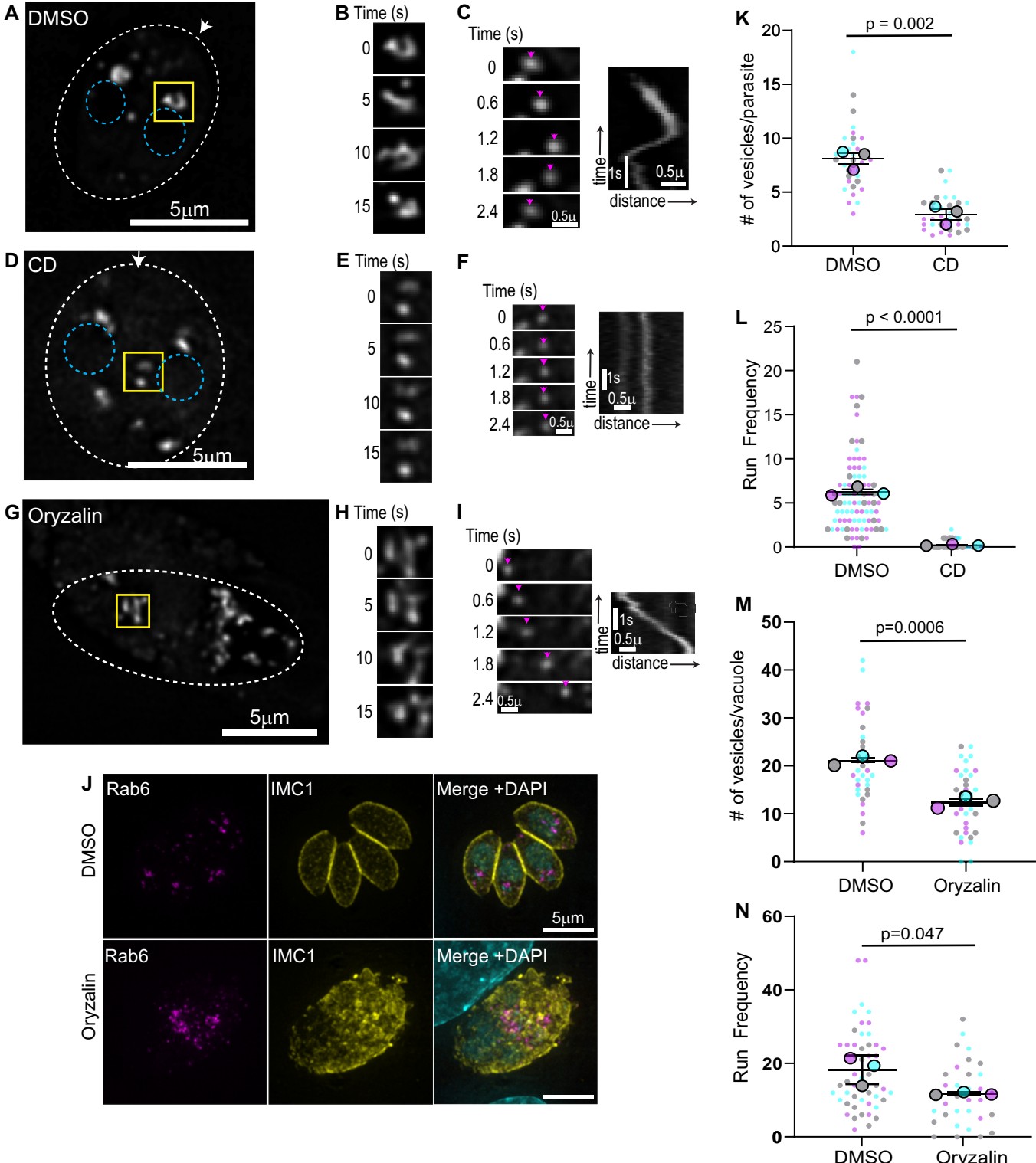

**Fig 3. Rab6 vesicle transport is actin dependent.** (A, D &G) RH parasites expressing EmGFP-Rab6 treated with DMSO (A) or cytochalasin D (CD) (D) or oryzalin (G). Dashed oval indicates the PV surrounding a 2-parasite vacuole. Location of the nucleus is indicated by the blue circles. Area in the yellow box was used to make inset (panels B, E and H). Arrow indicates parasites apical end. (B, E and H) Rab6 compartment dynamics. Images taken at 5 second intervals. (C, F and I)Time lapse images (left) and kymograph (right) depicting Rab6 directed vesicle motion. No directed motion was observed after cytochalasin D treatment indicated by the vertical line in kymograph. (J) Immunofluorescence image of parasites expressing EmGFP-Rab6 (magenta) treated with DMSO or oryzalin for 15 hours. Parasites were fixed and stained with an anti-IMC1 antibody (yellow) and DAPI (cyan). (K) Number of Rab6 vesicles per parasite was quantified in DMSO and CD treated parasites. (L) Run frequency (defined as the number of directed runs/parasite/minute) observed in DMSO and CD treated

parasites. (M) Number of Rab6 vesicles per vacuole was quantified in DMSO and oryzalin treated parasites. (N) Run frequency observed in DMSO and oryzalin treated vacuoles. In (K-N) results are from three independent experiments. Mean from each independent experiment is indicated with large circles. These values were used to calculate the average (horizontal bar), standard error of the mean (error bars), and P value. Raw data is shown with smaller colored circles. Experiment 1 in magenta, experiment 2 in cyan, experiment 3 in grey.

Although the movement of Rab6 vesicles was actin dependent in *T. gondii*, in mammalian cells there is a coordination between the actin and microtubule cytoskeletons to control movement in the endo-lysosomal system [57] and in other eukaryotes Rab6 vesicles are associated with both kinesin and dynein motors [58–60]. Thus, we sought to determine if microtubules played a role in either Rab6 compartment dynamics or Rab6 vesicle transport (Fig 3G–3J). *T. gondii* tubulins are more similar to plant tubulins than to animal tubulins, therefore we depolymerized the microtubule cytoskeleton using the plant microtubule polymerization inhibitor, oryzalin [37,61,62]. Oryzalin treatment prevents the formation of microtubules in the growing daughter cells and disrupts the parasites' polarized cytoskeletal organization (Fig 3G–3J). When EmGFP-Rab6 expressing parasites were treated with oryzalin and imaged using live microscopy, we observed a dynamic Rab6 compartment as seen in the control (Fig 3G and 3H) and directed Rab6 vesicle movements (Fig 3I), although the number of vesicles per vacuole and the run frequency were both reduced by approximately 30% after microtubule depolymerization compared to controls (Fig 3M and 3N). It is difficult to discern if these changes are due to the large morphological differences that are induced after oryzalin treatment or if microtubules do play a minor role in vesicle motion.

### Creation of conditional TgMyoF-knockdown parasite line

Since dense granule transport is dependent on F-actin and TgMyoF [39], and Rab6 vesicle movement is largely actin dependent, we investigated if TgMyoF was also required for Rab6 vesicle transport or compartment dynamics. We previously used an inducible Cre-LoxP system to create a parasite line deficient in functional TgMyoF, however, depletion of TgMyoF protein levels after TgMyoF gene excision took ~48 hours [39,63]. Thus, we created an inducible TgMyoF knockdown (KD) parasite line using the auxin-inducible degradation system where TgMyoF protein levels could be rapidly reduced [64,65] (Fig 4A). The endogenous TgMyoF gene was C-terminally tagged with an AID-HA epitope to create a TgMyoF-mAID-HA parasite line (referred to subsequently as TgMyoF-AID). PCR of genomic DNA and western blot was used to confirm the accurate integration of this construct (Figs 4B and S2). Treatment of TgMyoF-AID parasites with indole-3-acetic acid (IAA) for four hours resulted in depletion of TgMyoF to undetectable levels (Fig 4B). TgMyoF depletion was independently confirmed by anti-HA immunofluorescence (IF) (Fig 4C). To ensure that this new TgMyoF-AID parasite line exhibited similar phenotypes compared to previously generated TgMyoF inducible knockout lines [39,44] we assessed the effect of TgMyoF depletion on apicoplast inheritance. Tir1 parental and TgMyoF-AID parasites were treated with ethanol or IAA for 15 hours and the number of apicoplast per parasites was quantified. No defect in apicoplast inheritance was observed in Tir1 parental lines treated with either ethanol or IAA or in TgMyoF-AID parasites treated with ethanol (S3 Fig), whereas ~40% of TgMyoF deficient parasites did not contain an apicoplast after 15 hours of IAA treatment, indicating that this independently generated parasite lined exhibits aberrant apicoplast inheritance as demonstrated previously (S3B and S3C Fig) [39,44].

### TgMyoF is required for apical positioning and structural integrity of the PGCs

We were intrigued by the observation that F-actin was required for the apical positioning of the Rab6 compartment. The PGCs have a well-defined position adjacent to the Golgi, yet we

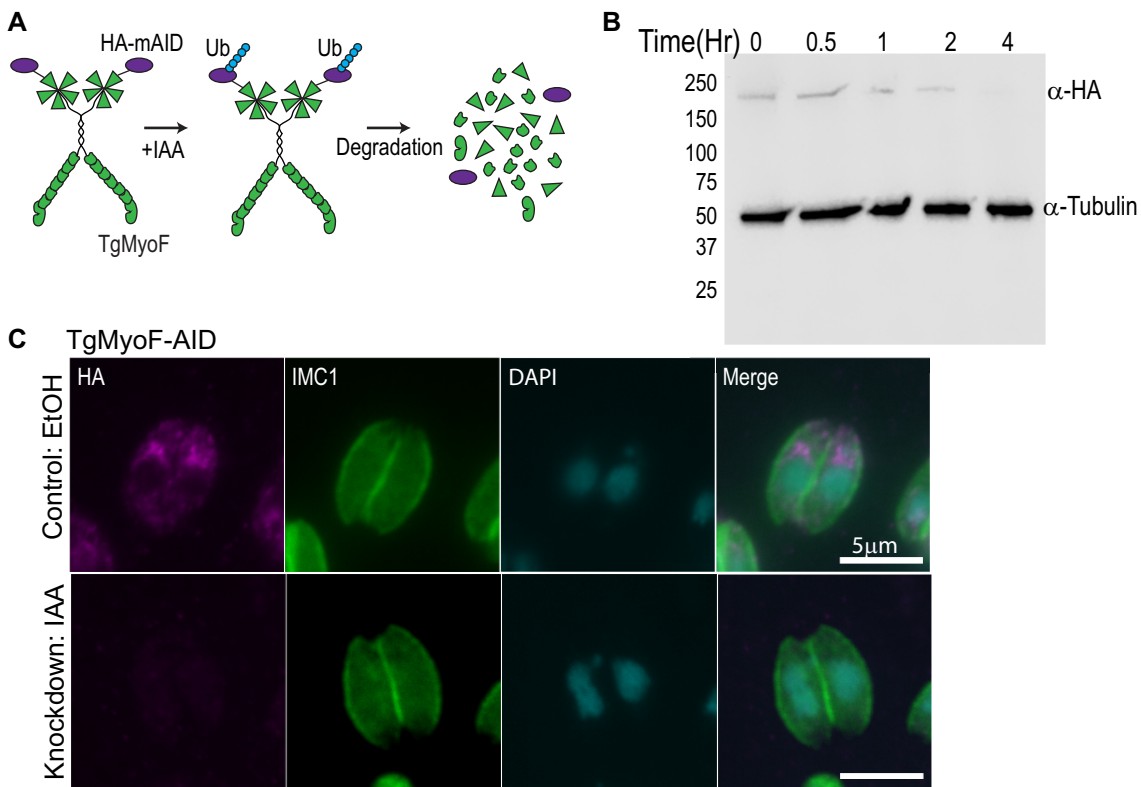

**Fig 4. Creation of TgMyoF-KD parasites using the auxin-inducible degradation system.** (A) Cartoon of TgMyoF knockdown strategy. (B) Western blot analysis of TgMyoF-AID-HA depletion as a function of IAA treatment time. Tubulin was used as the loading control. (C) Deconvolved epifluorescence images of TgMyoF-AID parasites treated with EtOH or IAA for 15 hours before fixation. Anti-HA immunofluorescence (magenta) was used to assess TgMyoF depletion. Anti-IMC1 antibody (green) and DAPI (cyan) staining used to visualize the parasite periphery and nuclei respectively.

have no mechanistic insight into how the position of these organelles is maintained. To determine if TgMyoF is required for the apical positioning of the PGCs, we ectopically expressed markers for these compartments, specifically EmGFP-Rab6, Neon-Rab5a, Neon-Rab7, Syn6-GFP and DrpB-GFP in TgMyoF-AID parasites and treated them with ethanol (EtOH; control) or IAA (to deplete TgMyoF) for 18 hours before fixation. In addition, EtOH and IAA treated parasites were fixed and stained with an anti-SORTLR antibody [53].As expected in control parasites, these compartments were positioned at the apical end of the parasite (Fig 5A left panels). After TgMyoF depletion however, the Rab5a, Rab6, DrpB, Syn6 and SORTLR compartments became fragmented and were found throughout the cytosol (Fig 5A; right panels). In order to determine how loss of TgMyoF affected the PGCs, we quantified compartment number in each parasite after ethanol or IAA treatment. In Rab5 and Rab6 expressing parasites we wanted to distinguish between the Rab6 or Rab5 compartment and Rab6/Rab5 vesicles. Therefore, vesicles which were typically smaller than 200μm$^2$ were excluded from this analysis. We found a statistically significant increase in the number of large objects stained with PGC markers after TgMyoF depletion in all cases (Fig 5B). In the case of Rab7, this protein had a diffuse localization in the cytosol after TgMyoF depletion (Fig 5A).

To confirm that these defects were due to loss of TgMyoF and not IAA treatment, Tir1 parental parasites were treated with either ethanol or IAA and scored as having either an "apical&intact" or a "fragmented&distributed" Rab6 compartment. In both cases, approximately ~25% of parasites were scored as having a fragmented Rab6 compartment, reflecting the

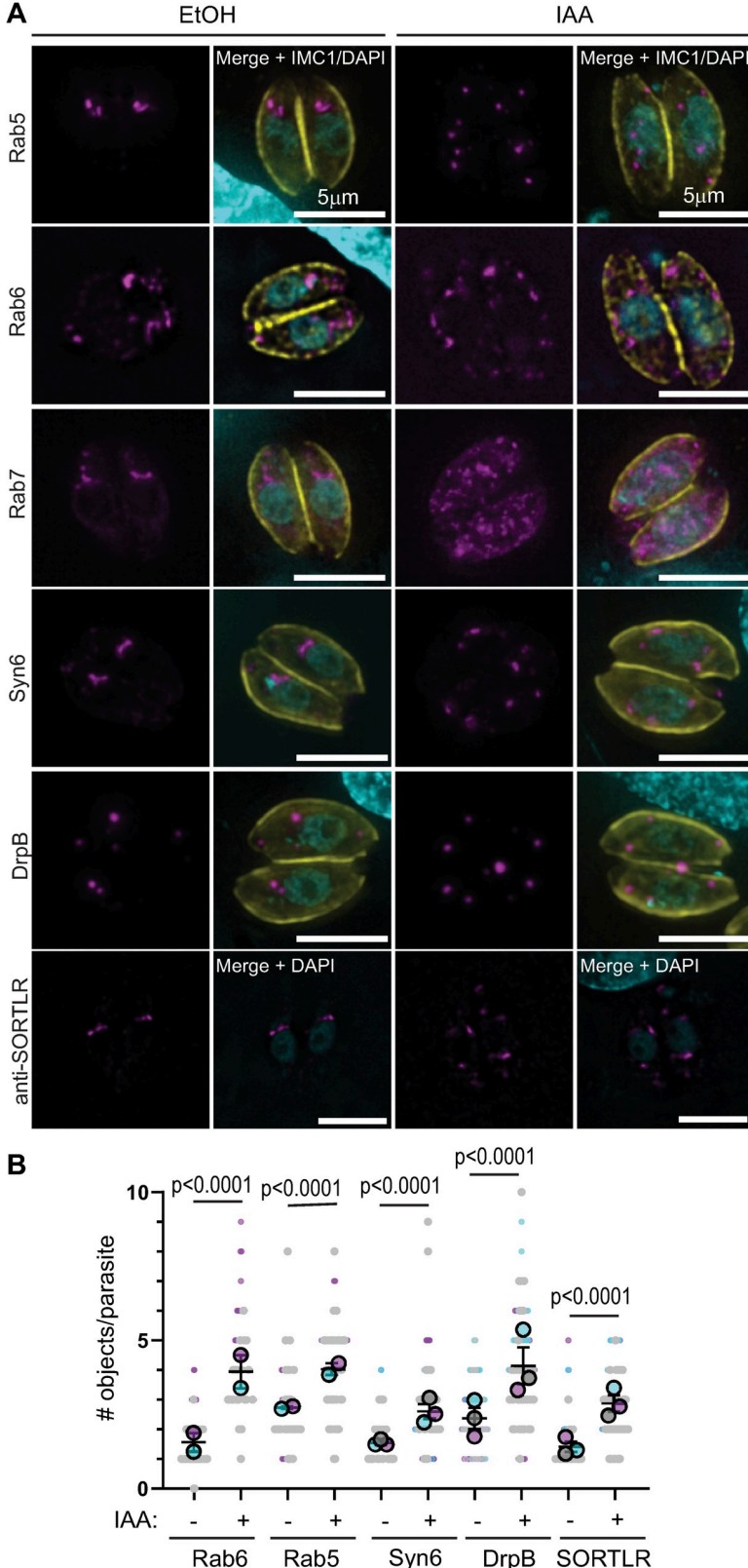

**Fig 5. TgMyoF knockdown results in fragmentation and dispersion of post-Golgi compartments.** (A) TgMyoF-AID parasites expressing Neon-Rab5, EmGFP-Rab6, Neon-Rab7, GFP-Syn6, GFP-DrpB (magenta), or

untransfected parasites, were treated with EtOH (left panels) or IAA (right panels) for 18 hours before fixation. Parasites were stained with anti-IMC1 to highlight the parasite periphery (yellow) and DAPI to stain DNA (cyan). Untransfected parasites were stained with anti-SORTLR antibody (magenta; lower panel). Scale bar is 5µm. (B) For each post-Golgi compartment, the number of objects was quantified in control and TgMyoF knockdown parasites. Combined results from two or three independent experiments. Mean from each independent experiment is indicated with large circles. These values were used to calculate the average (horizontal bar), standard error of the mean (error bars), and P value. Raw data is shown with smaller colored circles. Experiment 1 in magenta, experiment 2 in cyan, experiment 3 in grey.

percent of parasites at various stages of cell division, while the remaining ~75% of parasites contained an intact Rab6 compartment at the apical end of the parasite (S4A and S4B Fig).

Since Rab6 and Syn6 are localized to the same compartment in control parasites, we sought to determine if these proteins remained colocalized after compartment fragmentation. In TgMyoF deficient parasites expressing AppleFP-Rab6 and Syn6-GFP, we found that the co-localization between these proteins remained after compartment fragmentation (S5A Fig). In addition, 70% of parasites also contained Rab6+/Syn6- vesicles (S5A Fig, lower panel, white arrows). Next, we investigated if the Rab6 compartment, which are usually closely aligned with the trans-Golgi in control parasites, remained in close proximity to the trans-Golgi in TgMyoF depleted parasites. Therefore, we imaged EmGFP-Rab6 expressing parasites that were fixed and stained with an anti-SORTLR antibody, a marker of the trans-Golgi. These compartments are closely associated in ethanol-treated control parasites (S5B Fig). While some fragments remained adjacent after IAA treatment (solid arrowheads), this close association was not retained in most cases (S5B Fig; double arrowhead).

## TgMyoF plays a role in Rab6 vesicle transport

To assess the role of TgMyoF in the Rab6 compartment and Rab6 vesicle dynamics, TgMyo-F-AID parasites expressing EmGFP-Rab6 were treated with either EtOH or IAA for 18 hours before live-cell imaging. Similar to what was observed after actin depolymerization, loss of TgMyoF resulted in fragmentation of the Rab6 compartment (Fig 5A and 5B; left panels) and loss of this compartments dynamic reorganization (Fig 6A and 6B middle panel). There was also a decrease in the number of Rab6 vesicles exhibiting directed motion from 7.6±0.7 in the control to 3±0.45 in TgMyoF-KD parasites (Fig 6A and 6B right panels; and Fig 6C) (S3 Video). Although the number of directed runs was significantly reduced, vesicle velocities were the same in the absence of TgMyoF compared with controls (Fig 6D) (S1 Table).

These data combined with published work demonstrate that TgMyoF and actin are required for dense granule and Rab6 vesicle transport [39], apical positioning of the post-Golgi compartments, and inheritance of the apicoplast [44]. All of these organelles are part of the endomembrane network in *T. gondii*. Therefore, we wanted to determine if other organelles in this pathway, namely the ER, the Golgi, the micronemes and the rhoptries, relied on this acto-myosin system for their dynamics and/or morphology.

## TgMyoF-knockdown affects movement of ER tubules

The ER is a large membrane-bound organelle that has three distinct functional domains, the nuclear envelope, peripheral tubules and peripheral cisternae which form an extensive and continuous network in the parasite cytosol (Fig 7A) [52,66]. Live cell imaging of RH parasites with a fluorescently labeled ER, achieved by expression eGFP-SAG1ΔGPI-HDEL [54] (referred to subsequently as GFP-HDEL) reveals that the ER tubules are highly dynamic and undergo continuous reorganization (S4 Video). ER tubule rearrangements are clearly evident when the first frame of the movie was overlaid with images taken after 5, 10 and 15 seconds of

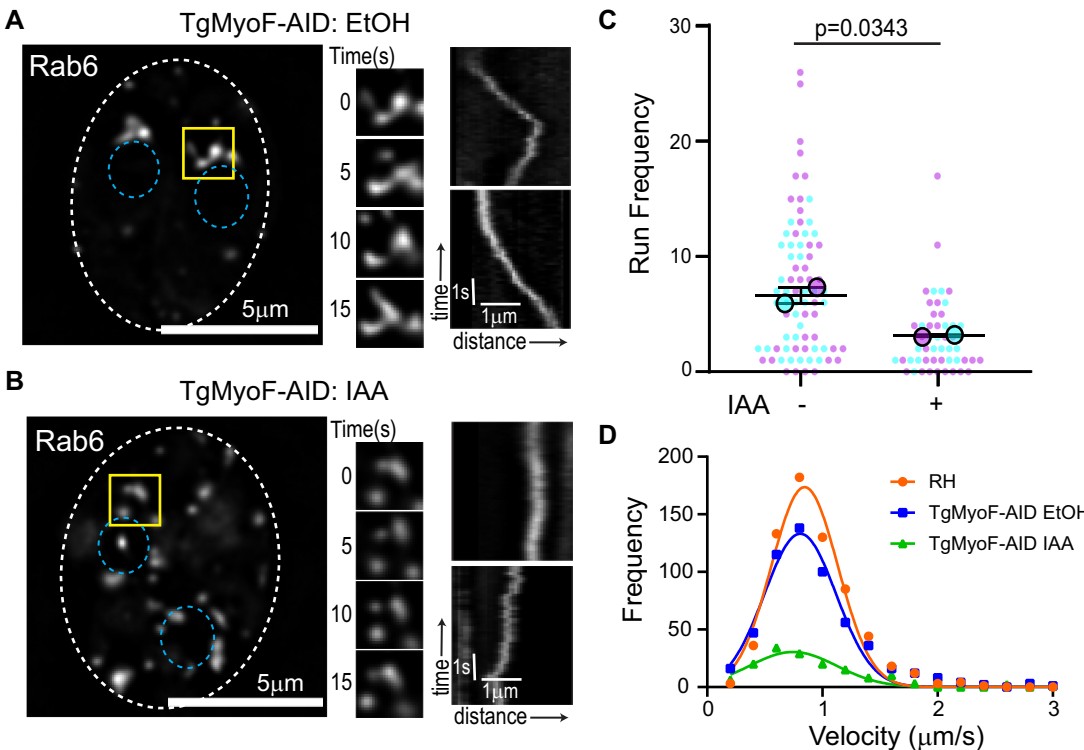

**Fig 6. TgMyoF is required for Rab6 vesicle transport.** (A-B) *Left panel*: TgMyoF-AID parasites expressing EmGFP-Rab6 treated with EtOH (A) or IAA (B) for 18 hours before imaging. Dashed oval indicates the PV surrounding a 2-parasite vacuole. Location of the nucleus is indicated by the blue circles. Area in the yellow box was used to make inset (middle panel). *Middle panel*: Images of the Rab6 compartment taken at 5 second intervals. *Right panel*: Kymograph depicting Rab6 vesicle motion. (C) Run frequency (# of directed runs/parasite/minute) of Rab6 vesicles in control and TgMyoF depleted parasites. Combined results from two independent experiments. Mean from each independent experiment is indicated with large circles. These values were used to calculate the average (horizontal bar), standard error of the mean (error bars), and P value. Raw data is shown with smaller colored circles. Experiment 1 in magenta, experiment 2 in cyan. (D) Frequency distribution of Rab6 vesicle velocities in RH parasites (orange) and in TgMyoF-AID parasites treated with EtOH (blue) or IAA (green) for 18 hours.

imaging (Fig 7A; middle panel) and when line scans were used to depict the fluorescence intensity at each of these time points (Fig 7A; right panel). ER tubule motility has been described previously in mammalian cells [67]. In this case, tubule dynamics are dependent on the microtubule cytoskeleton [68–70]. To determine if actin or microtubules were required for ER tubule movement, we treated RH parasites expressing GFP-HDEL with oryzalin for 15 hours or with CD for 30 minutes. After microtubule depolymerization, ER tubules spread through the parasite "masses" and these tubules remained motile (Fig 7B and S4 Video). After actin depolymerization, the ER tubule network appears less extensive and resulted in a dramatic loss of motile ER tubules as seen in time overlay images and line scans (Fig 7C and S4 Video). Since ER tubule motility was actin dependent, we investigated if the loss of TgMyoF also affected this motility. As seen previously with RH parasites, TgMyoF-AID parasites expressing GFP-HDEL treated with ethanol exhibited highly motile ER tubules. Numerous instances of tubule fusion and fission were observed over the 60 second imaging period (Fig 7D and S5 Video). When TgMyoF-AID parasites expressing GFP-HDEL were imaged after IAA treatment, we again observed a decrease in ER tubule motility as depicted with time overlay images and line scans, although there did not appear to be a collapse of the peripheral ER tubules (Fig 7E and S5 Video).

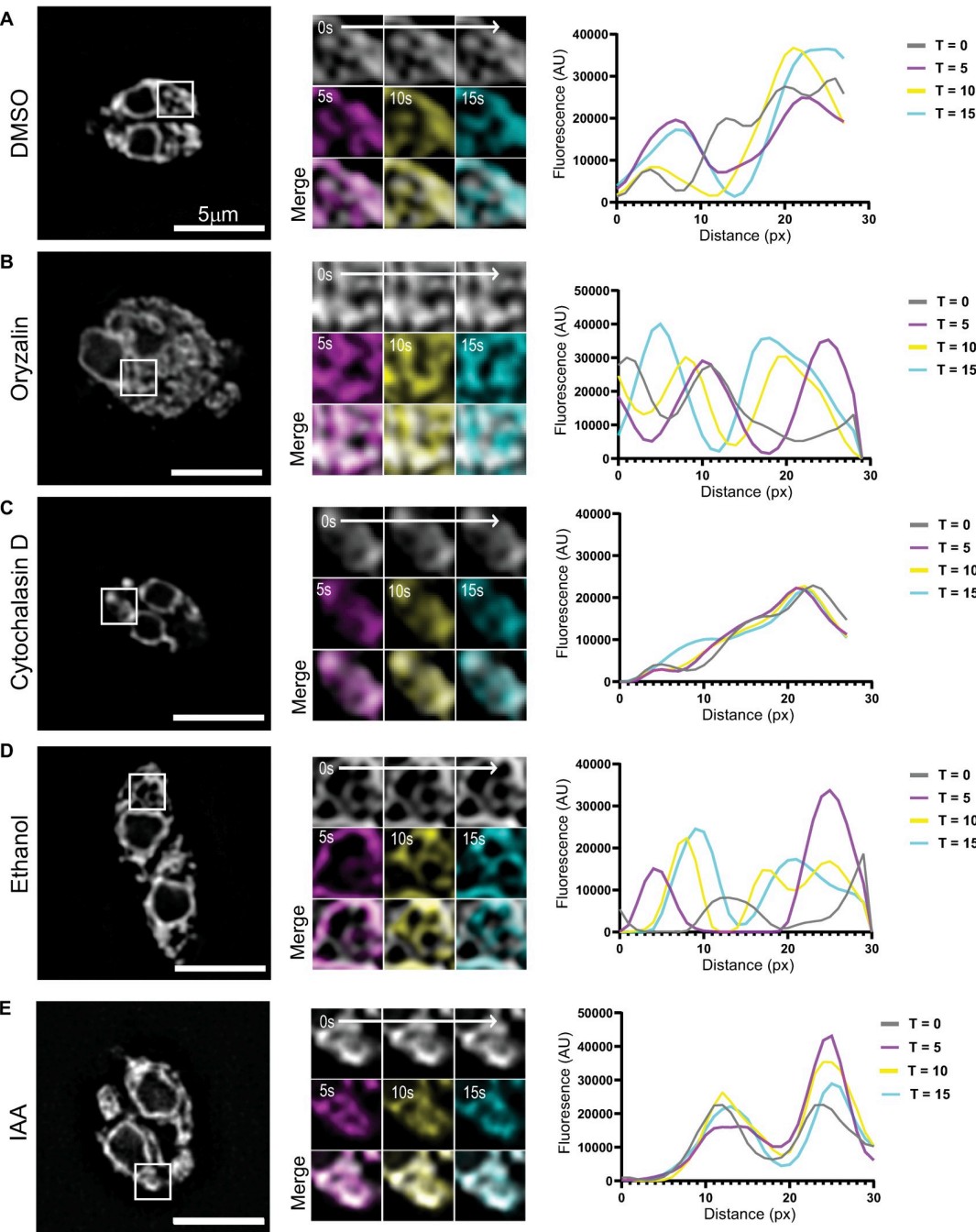

**Fig 7. TgMyoF is required for ER tubule movement.** (A-C) *Left panel*: RH parasites treated with DMSO, oryzalin for 18 hours or CD for 30 minutes. *Center panel*: magnification of area in the white box. First image of each movie (grey) was overlaid with images taken after 5s (magenta), 10s (yellow) and 15s (cyan) of imaging. *Right panel*: Line scan analysis of insets at 0s (grey), 5s (magenta), 10s (yellow) and 15s (cyan) time points. (D & E) *Left panel*: TgMyoF-AID parasites treated with EtOH (D) or IAA (E) for 15 hours where the ER is fluorescently labeled with eGFP. *Center panel*: magnification of area in the white box. First image of each movie (grey) was overlaid with images taken after 5s (magenta), 10s (yellow) and 15s (cyan) of imaging. *Right panel*: Line scan analysis of insets at 0s (grey), 5s (magenta), 10s (yellow) and 15s (cyan) time points.

## Actin depolymerization and TgMyoF depletion results in Golgi fragmentation

Next, we investigated if the loss of TgMyoF or actin depolymerization affected Golgi morphology. The cis-Golgi was fluorescently labeled by expressing Grasp55-mCherryFP in parental Tir1 or TgMyoF-AID parasites. Parasites were treated in EtOH or IAA for 15 hours. In Tir1 parasites treated with EtOH or IAA, and in TgMyoF-AID parasites treated with EtOH, 80% of parasites contained a single Golgi localized at the apical end of the nucleus while 20% of parasites were undergoing cell division and contain two Golgi per parasite as expected (Figs 8A, 8B, 8D and S4A–S4C). When TgMyoF-AID parasites were treated for 15 hours with IAA, only 25% of parasites contained a single Golgi, 52% contained two Golgi and 21% contained three Golgi (Fig 8B–8D). Similarly, actin depolymerization with cytochalasin D also resulted in an increased number of Golgi per parasite (Fig 8C and 8D). While the majority of Golgi remain closely associated with the nucleus after TgMyoF-knockdown or CD treatment, the apical positioning of the Golgi was lost. After TgMyoF depletion or actin depolymerization, 52% and 40% of parasites respectively contained Golgi in both the apical and basal ends of the parasites compared to just 5% of control parasites (Fig 8E).

Since the Golgi in *T. gondii* divides by binary fission during cell division [51,71], the increased number of Golgi observed after the loss of F-actin and TgMyoF could be due to the uncoupling of the Golgi division cycle from the cell cycle. To investigate the timing of Golgi fragmentation, TgMyoF-AID parasites expressing Grasp-mCherryFP were grown for 18 hours in the absence of drug, IAA was then added to parasites every 30 minutes for a total of 7 hours (Figs 8F–8H and S6A and S6B). The Golgi number began to increase within the first 90 minutes after IAA addition, even though TgMyoF was still detectable in parasites at this time (Fig 4). At the 4-hour time point, the time at which TgMyoF is completely depleted and a length of time shorter than one parasite division cycle, almost 60% of parasites had 2 Golgi, the same percentage that was observed at the 15-hour time point. This data indicates that Golgi fragmentation occurs quickly upon TgMyoF depletion but does not continue to fragment with extended IAA treatment times (Fig 8D–8F). Next, we investigated if the loss of TgMyoF affected the number of centrosomes per parasite as it had previously been demonstrated that centrosome duplication and Golgi fission are the first events to take place at the start of parasite division [52] (Fig 8A). TgMyoF-AID parasites were transfected with Grasp55-mCherry and centrin1-GFP [72] to label the Golgi and centrosomes respectively, and then treated with EtOH or IAA for 15 hours. In control parasites, 73% of parasites contained one Golgi and one centrin (1G/1C) while the remaining ~30% of parasites were at various stages of division and contained either one Golgi and two centrin (1G/2C) or two Golgi and two centrin (2G/2C) (Fig 9A and 9B). In contrast, only 20% of TgMyoF-KD parasites contained one Golgi and one centrin (1G/1C), while 41% contained two Golgi and one centrin (2G/1C), compared to just 3% of controls. 7% and 16% of IAA treated parasites contained one centrin and three Golgi (1C/3G) or two centrin and three Golgi (2C/3G) respectively, which were phenotypes that were never observed in the control parasites (Fig 9A and 9B). The number of centrosomes per parasite remained unchanged in TgMyoF-KD parasites compared to controls (Fig 9C). Thus, we conclude that TgMyoF and actin are important for controlling both Golgi number and apical positioning in interphase parasites and these phenotypes are not due to a block in the cell division cycle after centrosome duplication and Golgi division, the earliest events in the parasites cell division cycle [52].

Our data indicates that loss of TgMyoF, results in fragmentation of both the Golgi and the post-Golgi compartments. Since these compartments are highly interactive, we sought to determine if one of these effects was a down-stream consequence of the other. To investigate

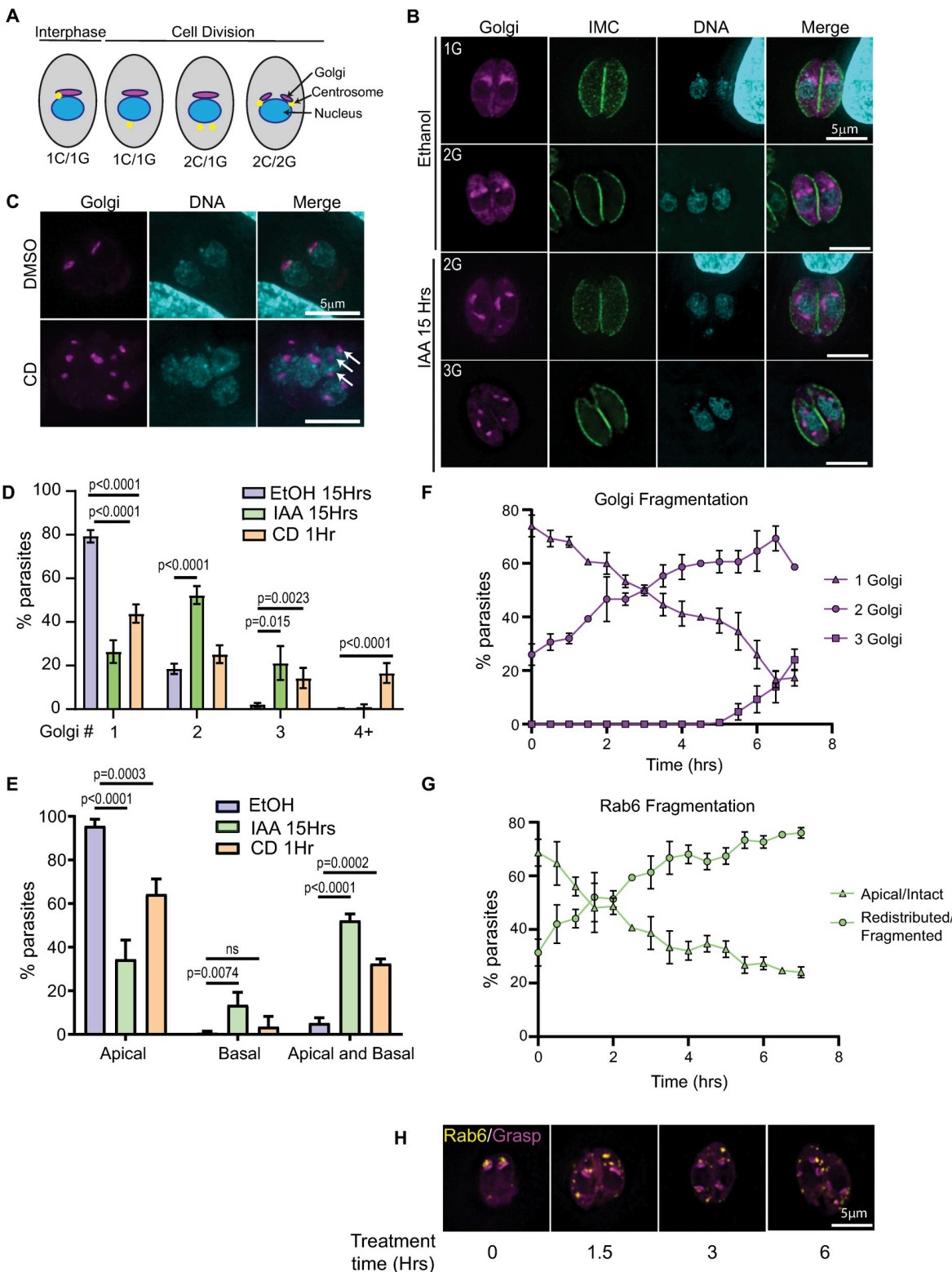

**Fig 8. TgMyoF and actin control Golgi morphology.** (A) Diagram of centrosome and Golgi division. (B) TgMyoF-AID parasites expressing Grasp55-mCherry (a marker of the cis-Golgi; magenta) were fixed and stained with anti-IMC1 (green) and DAPI (cyan) after 15 hours of EtOH or IAA treatment. (C) TgMyoF-AID parasites expressing mCherry-Grasp55 were treated with DMSO or CD for 60 minutes before fixation and DAPI staining. White arrows indicate Golgi fragments. (D) Quantification of the number of Golgi/parasite in TgMyoF-AID parasites treated with EtOH for 15 hours (blue), IAA for 15 hours (light green) or CD for 60 minutes (orange). Results are

combined data from at least 2 independent experiments. N>50 parasites/experiment. (E) Quantification of Golgi localization in apical only, basal only or both the apical and basal ends of the parasite. Results are combined data from 2 independent experiments. N>50 parasites/experiment. (F) In TgMyoF-AID parasites, the number of Golgi per parasite was quantified as a function of time after IAA addition (time 0). (G) In TgMyoF-AID parasites, the number of parasites with a fragmented and distributed Rab6 compartment was quantified as a function of time after IAA addition (time 0). (H) TgMyoF-AID parasites expressing EmGFP-Rab6 (yellow) and Grasp55-mCherry (magenta) treated with IAA for 0, 1.5, 3 and 6 hours.

this, we compared the timing of Rab6 compartment fragmentation to that of Golgi fragmentation (Fig 8F and 8G). The time course experiment was performed as described above with parasites expressing EmGFP-Rab6. Parasites with a fragmented Rab6 compartment were observed as early as 1.5 hours after IAA addition, with Rab6 compartment fragmentation seen in ~70% of parasites at the four-hour time point (Figs 8G and 8H and S6A and S6C). The fragmentation of the Rab6 compartment and the Golgi are concurrent and it appears that the fragmentation of one compartment is not an indirect effect of fragmentation of the other.

## TgMyoF depletion affects apical positioning of the rhoptries and Rop1 vesicle movement

It has been demonstrated previously that loss of TgMyoF results in the accumulation of intact micronemes and rhoptry proteins in the parasite's residual body [44]. In our independently generated TgMyoF conditional knockdown parasite line, we observe a similar phenotype. After 15 hours of IAA treatment, 73% of TgMyoF deficient vacuoles contained rhoptries

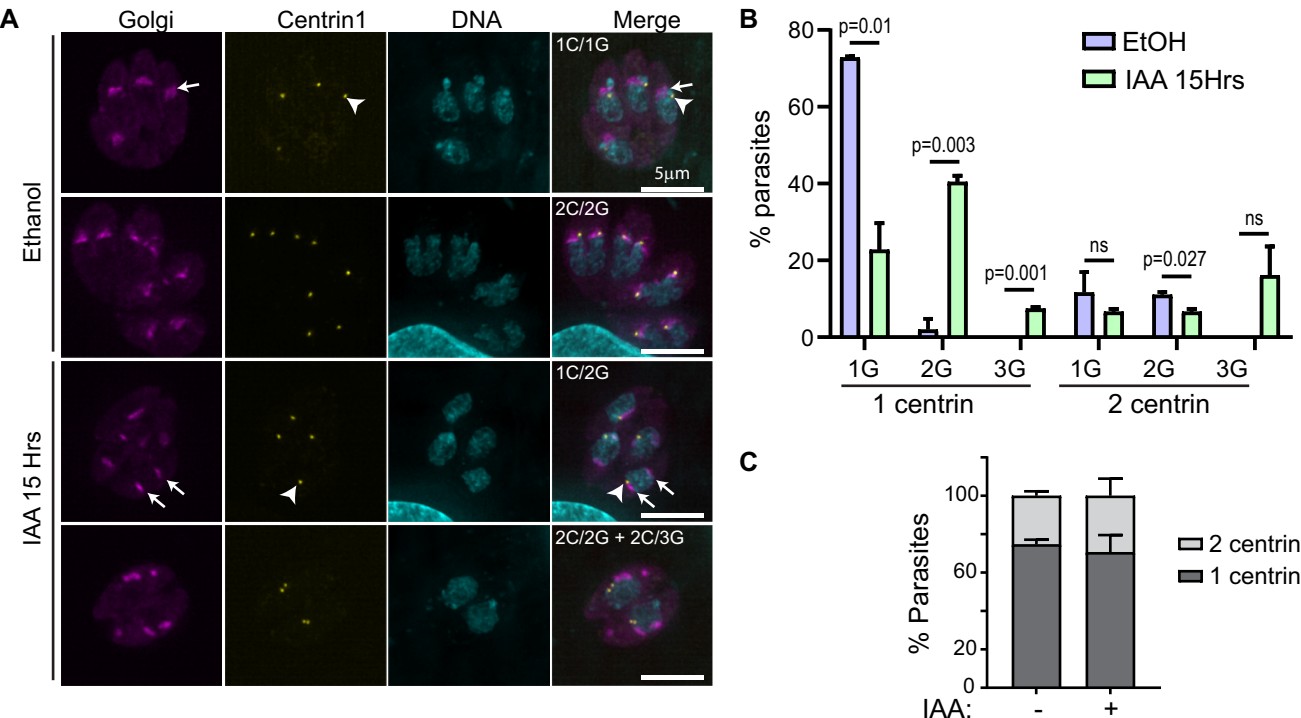

**Fig 9. TgMyoF knockdown does not affect centrosome number.** (A) TgMyoF-AID parasites expressing Grasp55-mCherry (magenta) and centrin 1-GFP (as a marker of the centrosome; yellow) were treated with EtOH and IAA for 15 hours then fixed and stained with DAPI (cyan). Arrowhead indicates the centrosome. Arrows indicate the Golgi. (B) Quantification of centrosome and Golgi number in control and TgMyoF depleted parasites. Combined results from 2 independent experiments. N = 77 parasites for control, N = 93 parasites for IAA treated. (C) Centrosome number per TgMyoF-AID parasite treated with EtOH or IAA. Combined results from 2 independent experiments. N = 77 parasites for control, N = 93 parasites for IAA treated.

(visualized by expression of Rop1-NeonFP) in the residual body compared to just 14% of controls. While 67% of TgMyoF deficient vacuoles exhibited accumulation of micronemes in the residual body (visualized using an anti-AMA1 antibody) compared to 5% of controls (Fig 10A and 10B). Despite the accumulation of these organelles in the residual body, the apical positioning of the micronemes was not affected in TgMyoF-knockdown parasites when assessed by IFA (Fig 10A). In contrast, there appeared to be an increased Rop1 fluorescence throughout the parasite cytosol (Fig 10A; magenta arrow). To further investigate the effects of TgMyoF depletion on rhoptry dynamics, we expressed Rop1-NeonFP in TgMyoF-AID parasites treated for 18 hours with either EtOH or IAA and imaged the parasites using live cell microscopy. In control parasites, the rhoptries were localized as expected at the apical end. The rhoptries were surprisingly dynamic, and like the Rab6 compartment, were continually rearranged and separate Rop1 positive compartments were observed to undergo both fission and fusion (Figs 10C inset and S1C and S6 Video). In addition, Rop1 vesicles were observed throughout the parasite and exhibited directed, motor-driven motion (S6 Video). As with the other vesicle types, we observe Rop1 vesicles moving towards both the apical and basal poles of the parasites and in some cases budding from the apical, presumably mature rhoptries (S1D Fig). Upon TgMyoF knockdown we observed a large decrease in the number of directed runs exhibited by Rop1--NeonFP vesicles from 11±1.1 in control parasites to 1.4±0.2 after IAA treatment, even though the total number of Rop1 vesicles was not statistically different between control and TgMyoF depleted parasites (6.25±0.4 and 9.1±0.6 in EtOH and IAA treated cells respectively) (Fig 10E and 10F). To further investigate the effect of TgMyoF knockdown on the apical positioning of the rhoptries, we compared Rop1-NeonFP fluorescence intensity at the apical and basal ends in control and TgMyoF depleted parasites. In control parasites the apical:basal ratio was 5.8 ±0.7, indicating a strong enrichment of Rop1-NeonFP at the parasites apical end. By comparison the apical:basal ratio in IAA treated parasites was only 2.1±0.15. While Rop1 is still enriched at the parasite's apical end, there is an increase in Rop1-NeonFP fluorescence at the basal end of TgMyoF-KD parasites compared to controls (Fig 10G).

## Discussion

The polarized endomembrane system in *T. gondii* is vitally important for the accurate trafficking of secretory proteins to the micronemes, rhoptries and dense granules. Our data demonstrate that F-actin and TgMyoF control the dynamics, positioning and morphology of the endomembrane network in *T. gondii*.

Actin-driven ER tubule motility observed in *T. gondii* is reminiscent of the microtubule-controlled ER tubule movements that have been described in mammalian cells [68,69,73] and actin/myosin XI driven tubule movements in plants [74]. The function of ER tubule movement is best understood in mammalian cells, where ER tubules form contacts with numerous organelles and regulates the timing and site of mitochondrial and endosome fission, as well as facilitating lipid and calcium exchange between organelles [75–77]. Future work is required to elucidate the importance of ER tubule movement in *T. gondii* biology.

The Golgi in *T. gondii* is localized adjacent to the nuclear envelope at the parasites apical end, appearing as a single stack when imaged by conventional fluorescence microscopy. Early in the parasite's cell division cycle, the Golgi elongates and divides by binary fission [52,71]. A second round of Golgi division then occurs, and two Golgi are inherited by each daughter [51] which appear to coalesce to form a single Golgi stack. However, it remains undetermined whether the two Golgi stacks undergo membrane fusion or if they are simply maintained in close proximity such that the stacks cannot be resolved using diffraction-limited microscopy. In this study, we quantified Golgi number in the first seven hours after IAA addition and one

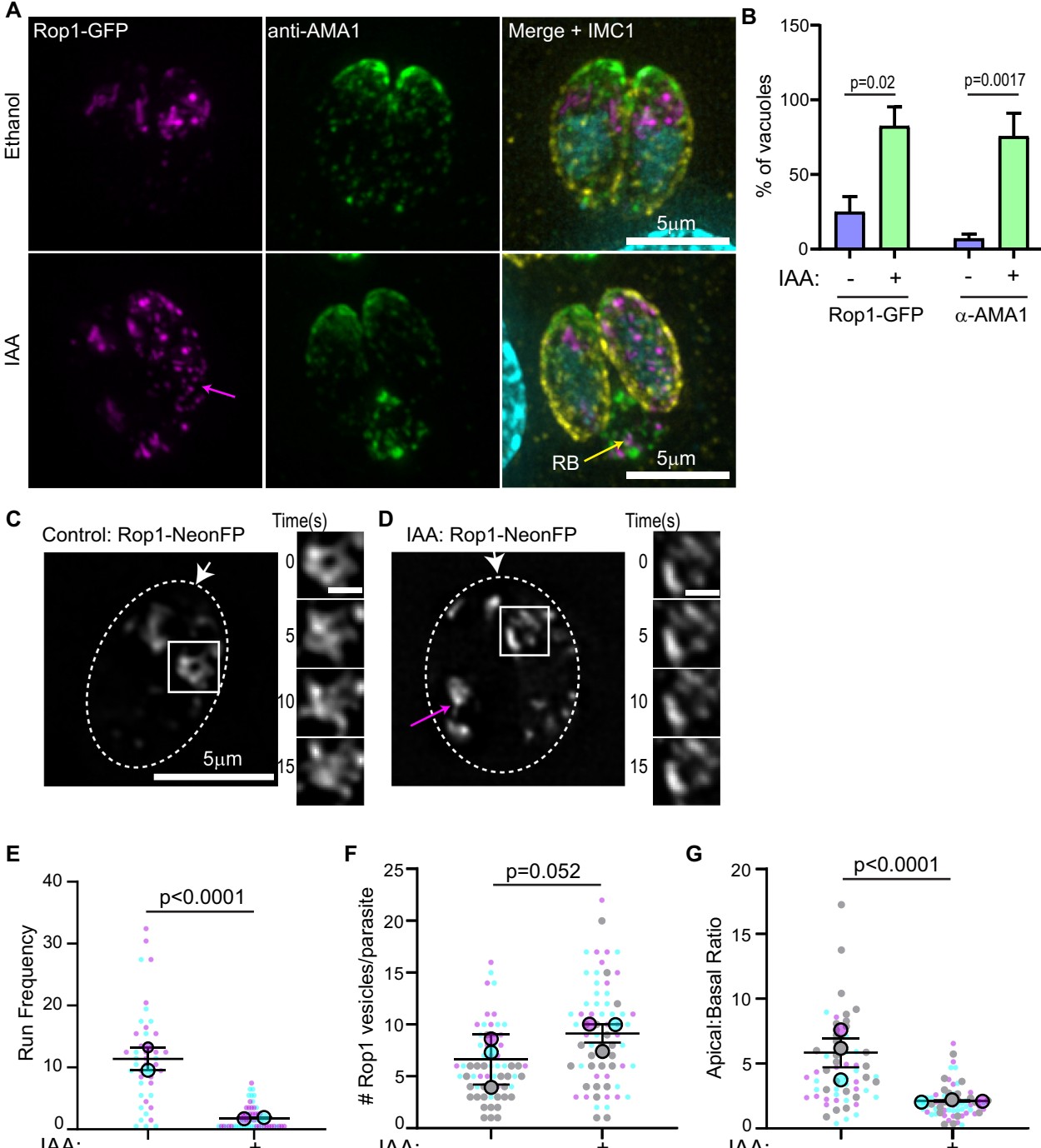

**Fig 10. TgMyoF depletion affects apical positioning of the rhoptries and Rop1 vesicle movement.** (A) TgMyoF-AID parasites expressing Rop1-NeonFP (magenta) were treated with EtOH and IAA for 15 hours before fixation and staining with an anti-AMA1 (green) and anti-GAP45 (yellow) antibodies and DAPI (cyan). Magenta arrow indicates Rop1 vesicles in the cytosol. Yellow arrow indicates the residual body (RB). (B) Quantification of TgMyoF-AID parasites with micronemes (visualized with anti-AMA1 antibody) and rhoptries (visualized by expression of Rop1-NeonFP) in the residual body. N = 57 vacuoles for AMA-1 quantitation, N = 77 vacuoles for Rop-1 quantitation from two independent experiments. (C&D) TgMyoF-AID parasites expressing Rop1-NeonFP were treated with EtOH or IAA for 15 hours before live cell imaging. *Right panel*: Rop1 localization. Magenta arrow indicates Rop1 mislocalization to the parasites basal end. White arrow indicates parasite apical end. Dashed oval indicates the PV in a 2-parasite vacuole. Area in the white box was used to make inset in the left panel. *Left panel*: Dynamics of rhoptries in the first 15 seconds of imaging. (E) Run frequency (# of directed runs/parasite/minute) of Rop1 vesicles in control and TgMyoF depleted parasites. See S1 Table for more details. (F) Number of Rop1 vesicles per parasite. N = 63 and 76 parasites in control and TgMyoF-KD parasites respectively in 3 independent experiments. (G) Ratio of Rop1 fluorescence in the apical and basal ends of the parasite. N = 63 and 76 parasites in control and

TgMyoF-KD parasites respectively in 3 independent experiments. (E-G) Mean from each independent experiment is indicated with large circles. These values were used to calculate the average (horizontal bar), standard error of the mean (error bars), and P value. Raw data is shown with smaller circles; experiment 1 in magenta, experiment 2 in cyan, experiment 3 in grey.

hour after actin depolymerization with cytochalasin D. Golgi fragmentation had occurred within each of these time frames, with no accompanying effect on centrosome number suggesting the changes in Golgi number are not due to a perturbation of the parasites cell division cycle. One possible explanation for these results is that the Golgi may be comprised of a paired stack held together in very close proximity in an actin-dependent manner. A similar observation was recently made by Kondylis and colleagues who demonstrated in Drosophila S2 cells that the Golgi consists of duplicated structural units that are separated during the G2 phase of the cell cycle, prior to mitosis [78]. Golgi separation could be artificially induced upon actin depolymerization with either cytochalasin D or latrunculin B. The observation that Golgi number increases upon IAA addition but then plateau's after ~6 hours suggests that the number of structural units that can be formed from a single Golgi is finite. In addition to the increase in Golgi number, many Golgi failed to maintain their position at the parasite's apical end and were observed associated with the lateral and basal sides of the nucleus (Fig 8B–8E). While centrosome number was not affected by the loss of TgMyoF or actin, previous data [44] demonstrates that centrosome position is affected by the loss of TgMyoF, it is possible that this change in centrosome position contributes to the change in Golgi position or morphology observed in TgMyoF knockdown parasites. Collectively, we conclude that TgMyoF and actin are important for controlling both Golgi number and apical positioning in interphase parasites and these phenotypes are not due to a block in the cell division cycle after centrosome duplication and Golgi division, the earliest events in the parasites cell division cycle [52].

After exiting the Golgi, proteins destined for the micronemes and rhoptries are trafficked to one or more post-Golgi compartments. We have identified Rab6 as a new marker of the syntaxin6 compartment. Syntaxin6 is a protein previously shown to play a role in retrograde trafficking from the Rab5a and Rab7 compartments to the Golgi [25]. Our data builds upon previously published results indicating that Rab6 localizes to cytosolic vesicles and a compartment at the apical end of the nucleus, thought previously to be the Golgi since TgRab6 localizes to the Golgi when heterologously expressed in mammalian cells [25]. However, we find no evidence that Rab6 is found in the cis- or trans-Golgi. Additionally, while we also observe Rab6 vesicles throughout the parasite, Rab6 does not colocalize with a marker for the dense granules, indicating these are distinct vesicle types. The discrepancy between our results and previously published data, is likely due to the unavailability of organelle markers at the time of the previous publication [25]. The function of Rab6, and the "destination" of Rab6 vesicles requires further investigation. In mammalian cells, Rab6 vesicles are formed at the trans-Golgi and control both anterograde trafficking to the plasma membrane and retrograde trafficking to the ER [60,79,80]. While we observe Rab6 vesicles moving in both the anterograde and retrograde directions, the vesicles are frequently observed to switch directions and anterograde movement doesn't necessarily indicate that the vesicle is being targeted to the plasma membrane or to an anterograde compartment. The speed of vesicle movement requires us to image at a high frame rate (10 frames per second) and therefore we are limited to imaging durations on the order of minutes and cannot ascertain from this imaging the ultimate destination of the Rab6 vesicles.

The Rab6 compartment is dynamic and has a tubular morphology that undergoes continuous rearrangement. We observed new tubules growing from the compartment while others retract. Vesicles were observed to bud from the tip of these tubules and subsequently exhibited

actin and TgMyoF dependent directed movement. This dynamic morphology closely mirrors the dynamics of endosomal compartments in mammalian cells. In this case, tubule formation is driven by the coordinated activity of microtubule motors [81], branched actin networks nucleated by WASH and the Arp2/3 complex [82,83], and BAR domain-containing proteins that induce membrane curvature [84]. No recognizable BAR-domain containing proteins or Arp2/3 complex proteins are found in *T. gondii*, [85,86] indicating that the molecular mechanisms underlying tubule formation in the Rab6 compartment are distinct from the mechanisms of tubule formation in mammalian cells and will require further investigation.

The apical localization of the Rab6/Syn6, Rab5a, Rab7 and DrpB compartments are all dependent on F-actin and TgMyoF. This dependence on TgMyoF and actin for apical positioning suggests that these post-Golgi compartments are associated with the actin cytoskeleton, or that the physical connections between these compartments, that maintain the compartments in close proximity at the apical end, are formed in an actin dependent manner. Future studies are needed to further identify the molecular players that control the associations between these compartments.

Formation of rhoptry organelles is cell cycle regulated and occurs in the mid to late stages of daughter cell development [31,52]. Upon exit from the Golgi, rhoptry proteins are trafficked through the Rab5 compartment and then processed in a premature-rhoptry from which the mature rhoptry develops [10,22]. Rhoptries had previously been shown to accumulate in the residual body upon actin depolymerization or TgMyoF knockdown, implicating TgMyoF in rhoptry trafficking and/or apical anchoring. Here, we further characterized the role of TgMyoF in rhoptry trafficking by imaging rhoptry dynamics in parasites expressing Rop1-NeonFP using live cell microscopy. As observed with the Rab6 compartment, the rhoptries undergo continual rearrangement and Rop1 vesicles were observed budding from the rhoptry bulb. Rop1 positive vesicles exhibited TgMyoF-dependent directed movement throughout the parasite cytosol. In the absence of TgMyoF, there was a significant decrease in the number of directed runs exhibited by Rop1-NeonFP vesicles. Currently, we do not know the function or subcellular destination of these rhoptry derived vesicles, these vesicles might represent the continual delivery of rhoptry cargo to the rhoptries in interphase parasites or the recycling of material from the rhoptries back to the VAC or Golgi. Although the rhoptries are formed during parasite division, these data suggest that acto-myosin dependent trafficking of proteins to or from the rhoptries occurs continuously. Further work will be required to answer the outstanding questions on rhoptry formation, trafficking and turnover.

There is an incomplete understanding of the mechanisms by which the mature rhoptries are anchored to the apical end of the parasite. In the absence of TgARO1, a membrane-associated rhoptry protein, the apical positioning of the rhoptries was lost completely [87]. This contrasts with the TgMyoF knockdown phenotypes where there is increased accumulation of rhoptry marker protein throughout the parasite cytosol, shown by an increase in Rop1 fluorescence at the parasites basal end (Fig 10G), even though most parasites retain at least some intact rhoptries at the apical end. Thus, the TgARO1 knockdown parasites have a more severe rhoptry localization and morphology defect than TgMyoF knockdown parasites. Although TgMyoF was shown to interact indirectly with TgARO1 [87], the differences in the severity of these phenotypes suggest that TgMyoF is not required for TgARO1 anchoring activity. Our data suggests that TgMyoF is required for movement of immature rhoptries to the apical tip but is not required for TgARO1-dependent anchoring once the organelles have reached their destination.

This study demonstrates that loss of TgMyoF alters the dynamics, positioning, and movement of a wide array of organelles in the endomembrane pathway in *T. gondii*. Future studies will be required to elucidate the mechanistic underpinnings of these phenotypes. It will be

important to determine if TgMyoF interacts directly with some or all of these organelles or if some of these phenotypes are downstream consequences of disruption to intracellular trafficking caused by the loss of TgMyoF. Given the structural similarity between TgMyoF and the well characterized cargo transporter myosin V, we previously hypothesized that TgMyoF bound dense granules via its C-terminal WD40 domain and transported cargo by moving processively on filamentous actin [39]. The large number of membrane-bound organelles whose movement is affected by the loss of TgMyoF makes elucidating TgMyoF's mechanism of action even more pertinent. Outstanding questions include: Does TgMyoF associate directly with each membrane-bound organelle? If so, what is the molecular basis of this association? How do these molecular complexes vary for each cargo and how are these interactions regulated? Does TgMyoF have the capacity to transport cargo as either a single motor or an ensemble? Future work aimed at identifying TgMyoF interacting proteins and modes of regulation will provide new insight into mechanisms of cargo transport in *T. gondii*.

## Materials and methods

### Cell culture and parasite transfection

*T. gondii* tachyzoites derived from RH strain were used in all experiments. Parasites were maintained by continuous passage in human foreskin fibroblasts (HFFs) in Dulbecco's Modified Eagle's Media (DMEM) (ThermoFisher, Carlsbad CA) containing 1% (v/v) heat inactivated fetal bovine serum (FBS) (VWR, Radnor PA), 1X antibiotic/antimycotic (ThermoFisher) as previously described [88]. Parasites were transfected as described previously [88] using a BTX electroporator set as follow: voltage 1500V; resistance 25Ω and capacitance 25μF.

### Drug treatment

To determine the effect of actin depolymerization on endomembrane organization and dynamics, or endoplasmic reticulum tubule dynamics, transfected parasites were grown for 15–18 hours in confluent HFF monolayers, treated for 30–60 minutes with either 2μM cytochalasin D or equivalent volume of DMSO before live-cell imaging in the presence of drug as described below. To determine the effect of microtubule depolymerization on Rab6 compartment dynamics/vesicle transport and endoplasmic reticulum tubule dynamics, transfected cells were treated for 15–18 hours with either 2.5μM oryzalin or an equivalent volume of DMSO. Live-cell imaging was performed in the presence of 2.5μM oryzalin as described below.

To deplete TgMyoF, TgMyoF-AID parasites were treated with a final concentration of 500μM IAA, diluted 1:1000 from a 500mM stock made in 100% EtOH. For live-cell imaging experiments treated time ranged from 15–18 hours. For western blot, treatment time was varied as indicated in Fig 4.

### Construction of expression plasmids

A list of plasmids, primers and gene accession numbers used in this study can be found in S2, S3 and S5 Tables respectively.

**Creation of pTKOII-MyoF-mAID-HA.** pTKOII-MyoF-EmeraldGFP (EmGFP) [39] was digested with BglII and AflII to remove the EmGFP coding sequence. AID-HA was amplified by PCR using the AID-HA ultramer as a template and primer pairs AID-HA F and AID-HA R. Plasmid backbone and the PCR product was gel purified and ligated via Gibson assembly using NEBuilder HiFi DNA assembly master mix as per manufacturer's instructions (New

England BioLabs; Ipswich, MA). Plasmids were transfected into NEB5α bacteria and positive clones screened by PCR and verified by Sanger sequencing.

**Creation of pmin-eGFP-Rab6-Ble.** To create parental plasmid pmin-eGFP-mCherry-Ble, pmin-eGFP-mCherry was digested with KpnI and XbaI. pGra1-Ble-SAG1-3'UTR plasmid [89] was digested with KpnI and XhoI to remove the ble expression cassette. A fill-in reaction was performed to produce blunt ends by incubating plasmids with 100μM dNTPs and T4 DNA polymerase at 12˚C for 15 minutes. Digested plasmids were gel purified and ligated together using T4 DNA ligase (New England Biolabs). Plasmids were transfected into NEB5α bacteria and positive clones screened by PCR and verified by Sanger sequencing. To create pmin-EmGFP-Rab6-ble, pmin-eGFP-mCherry-Ble was digested with NheI and AflII to remove eGFP-mCherry sequence. EmGFP was amplified by PCR with EmGFP-R6F and EmGFP-R6R primer pairs using pTKOII-MyoF-EmGFP as a template. Rab6 coding sequence was amplified by PCR using RH cDNA as a template and Rab6F and Rab6R primers. Digested plasmid backbone, EmGFP and Rab6 PCR products were gel purified and ligated using NEBuilder HiFi DNA assembly master mix as per manufacturer's instructions (New England Bio-Labs). Plasmids were transfected into NEB5α bacteria and positive clones screened by PCR and verified by Sanger sequencing.

**Creation of ptub-SAG1-ΔGPI-HDEL.** Parental plasmid ptub-SAG1-ΔGPI [39] was digested with AflII. HDEL ultramers (S3 Table) reconstituted to a concentration of 200mM in duplex buffer (100mM K Acetate; 30mM Hepes pH 7.5), were combined in equal ratios and headed to 95˚C for 5 minutes before cooling slowly to room temperature. Duplexed ultramers were diluted 1:100 in molecular biology grade water and ligated to digested plasmid using NEBuilder HiFi DNA assembly master mix as per manufacturer's instructions (New England BioLabs). Plasmids were transfected into NEB5α bacteria and verified by Sanger sequencing.

**Creation of pmin-NeonFP-Rab5a and pmin-NeonFP-Rab7.** Rab5a and Rab7 coding sequences were amplified by PCR using primers sets Rab5F/R, Rab7F/R and pTg-HARab5a and pTg-HARab7 [27,90] as templates. NeonGreen was amplified by PCR using Neon-R5F/Neon-R5R, Neon-R7F/Neon-R7R primer sets and Ty1-NeonGreenPave as a template. pmin-EmGFP-Rab6 plasmid was digested with NheI and AflII to remove EmGFP-Rab6 coding sequence. Plasmid backbones and PCR products were gel purified and annealed using Gibson assembly with NEBuilder HiFi DNA assembly master mix as per manufacturer's instructions. Plasmids were transfected into NEB5α bacteria and positive clones screened by colony PCR and verified by Sanger sequencing.

**Creation of ptub-Rop1-NeonGreenFP.** ptub-SAG1-ΔGPI-GFP plasmid was digested with NheI and AflII to remove SAG1-GFP coding sequence. Rop1-GFP coding sequence was amplified using Rop1 F/R primer pairs and RH cDNA as a template. NeonGreen was amplified by PCR using NeonRop1F and NeonRop1R primer pairs. Plasmid backbones and PCR products were gel purified and annealed using Gibson assembly with NEBuilder HiFi DNA assembly master mix as per manufacturer's instructions. Plasmids were transfected into NEB5α bacteria and positive clones screened by PCR and verified by Sanger sequencing.

**Creation of TgMyoF-AID parasite line.** The pTKO2_MyoF_mAID-HA plasmid was linearized with SphI and 25μg was transfected into $1\times10^7$ $\Delta Ku80$:$\Delta HXGPRT$:*Flag-Tir1* parental parasites (a gift from Dr. David Sibley, Washington University [64]). Parasites were selected using mycophenolic acid (MPA) (25 μg/ml) and xanthine (50 μg/ml) until approximately 70% of the parasites were HA positive. Clonal parasite lines were obtained by limited dilution into a 96 well plate. After 7 days of growth, wells containing a single plaque were selected for further analysis. All HA positive clones were amplified in a 6 well plate and genomic DNA isolated using Qiagen DNeasy blood and tissue kit as per manufactures instructions (Qiagen, Germantown, MD) (Cat #69504). Genomic DNA was analyzed for correct insertion of the

pTKO2_MyoF_mAID HA plasmid into the TgMyoF genomic locus by PCR using primers listed in S3 Table as outlined in S2 Fig.

## Western blot

To assess the extent of TgMyoF depletion after auxin addition as a function of time, $8\times10^6$ extracellular TgMyoF-AID parasites were incubated in 500μl of DMEM containing 1% FBS, antibiotic/antimycotic and 500μM IAA and incubated at 37°C, 5% $CO_2$ for 0, 0.5, 1, 2, or 4 hours. At each time point parasites were centrifuged at 1200xg for 5 minutes and resuspend in 25ul of 1xPBS containing protease inhibitor cocktail (MilliporeSigma, St. Louis MO; Cat # P8340). 25ul of 2XLamelli buffer (Biorad, Hercules, CA; Cat # 1610737) containing 100mM DTT was added to each sample and boiled at 95°C for 10 minutes. Samples were run on 4–12% gradient gel (Biorad; Cat# 456–1064), transferred to PVDF for 60 minutes at 4°C at 100V using the Biorad blotting system. PVDF membranes were blocked overnight at 4°C in 5% milk, 1XTBS-T (150mM NaCl, 20mM Tris base, 0.1% tween20 (v/v) pH 7.4) before blotting with tubulin and HA primary antibodies and anti-mouse and anti-rat secondary antibodies. All antibodies were incubated with PVDF for 60 minutes at room temperature with 3x10-minute washes with 1xTBS-T in between antibody incubations. Immunoblots were developed using Pierce ECL western blotting substrate (ThermoFisher, Cat# 32209) and visualized using LiCor Odyssey Fc. Antibody dilutions and product information can be found in S4 Table.

## Microscopy

**Parasite transfection for live-cell imaging or immunocytochemistry.**  25μg of each plasmid was transfected as described above. Transfected parasites were grown for 15–18 hours in confluent HFF monolayers grown on either MatTek dishes (MatTek corporation, Ashland MA) or on coverslips before either live cell imaging or immunocytochemistry.

**Live-cell microscopy.**  Growth media was replaced with Fluorobrite DMEM (ThermoFisher; Cat# A19867) containing 1% FBS and 1x antimycotic/antibiotic pre-warmed to 37°C. Images were acquired on a Cytiva (formally GE Healthcare) DeltaVision Elite microscope system built on an Olympus base with a 100x 1.39 NA objective in an environmental chamber heated to 37°C. This system is equipped with a scientific CMOS camera and DV Insight solid state illumination module. Image acquisition speeds were determined on a case-by-case basis as noted in the video legends.

**Immunocytochemistry.**  Parasites were fixed with freshly made 4% paraformaldehyde (Electron microscopy sciences, Hatfield, PA; Cat# 15714) in 1xPBS (ThermoFisher; Cat# 18912–014) for 15 minutes at RT. Cells were washed three times in 1xPBS and permeabilized in 0.25% TX-100 diluted in 1xPBS for 10 minutes at room temperature before washing three times in 1xPBS. Cells were blocked in 2% BSA-1XPBS for 15 minutes before antibody incubations. All antibodies were diluted in 0.5% BSA-1xPBS at the concentrations indicated in S4 Table. DNA was stained with 10μM DAPI diluted in 1xPBS for 10 minutes and then washed three times in 1xPBS. Cells in mattek dishes were either imaged immediately or stored in 1xPBS at 4°C. Coverslips were mounted onto slides using either Prolong Gold anti-fade reagent (ThermoFisher; Cat # P36930) or Prolong Diamond anti-fade reagent (ThermoFisher; Cat #P36965) and allowed to dry overnight before imaging.

## Image analysis

**Effect of IAA on *ΔKu80:ΔHXGPRT:Flag-Tir1* parental parasites.**  To assess whether IAA treatment affected apicoplast inheritance, *ΔKu80:ΔHXGPRT:Flag-Tir1* parental parasites

were grown in 500μm IAA or an equivalent volume of ethanol for 18 hours. Immunocyto-chemistry staining was carried out as described above using DAPI, anti-Cpn60, and anti-IMC1 antibodies. The number of apicoplast per parasite was scored in 50 parasites, in each of three independent experiments.

To assess whether IAA treatment affected Golgi or Rab6 compartment fragmentation, *ΔKu80:ΔHXGPRT:Flag-Tir1* parental parasites were doubly transfected with ptub-GRASP-mCherry and pmin-EMGFP-Rab6 as described above. Parasites were grown in 500μM IAA or an equivalent volume of ethanol for 18 hours before paraformaldehyde fixation. The number of Golgi per parasite was counted in 50 parasites, in each of three independent experiments. The Rab6 compartment was scored as "apical & intact" or "fragmented & dispersed" in 50 parasites in each of three independent experiments.

**Vesicle tracking and counting.** Vesicle tracking was performed using MtrackJ plug-in in Fiji (National Institutes of Health) as previously described [56]. Vesicle motions were categorized as directed if vesicles moved continuously in one direction for at least 10 frames. During each 60 second imaging period individual vesicles can exhibit multiple modes of motion, i.e., transition between stationary, directed movement and diffusive-like motion. In order to obtain accurate velocity and run-length data, vesicles are only tracked while moving in a directed manner and these movements are described in the paper as "directed runs". Line scan analysis was performed using the "plot profile" tool in Fiji. Kymographs were made using Fiji plugin KymographBuilder. Fiji plugin cell counter was used to quantify the number of vesicles per parasites. Statistical significance was determined using students t-test.

**Effect of TgMyoF depletion on post-Golgi compartment number.** To count number of PGC "objects" in each parasite in control and TgMyoF knockdown parasites, transfected parasites were fixed and stained with the anti-IMC1 antibody and DAPI and z-stack images were acquired using Deltavision elite imaging system. For each PGC image, maximum intensity projection of Z-stacked images were converted to binary images using Fiji and particles counted using the "analyze particles" tool. Rab5 and Rab6 positive vesicles were excluded from this analysis by removing objects smaller than 200μm$^2$ from this analysis. IMC1 staining was used to determine number of parasites per vacuole. Statistical significance was determined using students t-test.

**Time course of Rab6 and Golgi fragmentation.** To assess the time-point after auxin addition at which the integrity of the Golgi body and the Rab6 compartment were affected, TgMyoF-AID parasites, doubly transfected with ptub-GRASP-mcherry and pmin-EMGFP-Rab6, were grown for 18 hours in confluent HFF monolayers on Labtek 8-well plates. At 30-minute intervals, growth media was removed and replaced with growth media supplemented with 500 μM IAA for 7 hours. These parasites were also grown in parallel with an equivalent volume of ethanol for the entire 7-hour time course. *ΔKu80:ΔHXGPRT:Flag-Tir1* parental parasites were doubly transfected with ptub-Grasp-mCherry and pmin-EmGFP-Rab6 and supplemented with either 500 μM IAA or an equivalent volume of ethanol for the entire 7 hour time course. Parasites were fixed with 4% freshly made paraformaldehyde 1xPBS for 15 minutes at RT. Cells were washed three times in 1xPBS before imaging. Number of Golgi per parasite and number of parasites containing intact-apical or fragmented-distributed Rab6 were individually assessed during the time course. Fifty parasites were scored at each 30-minute time point between 0 and 7 hours in three independent experiments.

**Quantification of Golgi and centrosome number.** To quantify the number of Golgi and centrosome per parasite, TgMyoF-AID parasites were transfected with Grasp55-GFP plasmid alone or Grasp55-mCherry and Centrin1-GFP plasmids, grown in confluent HFF monolayers overnight and treated with either EtOH or IAA for the final 15- hours of growth before fixation and processing for immunofluorescence. Number of Golgi per parasite and number of

centrosomes per parasites were counted manually using the cell count tool in Fiji. Statistical significance was determined using students t-test.

**Quantification of rhoptry and micronemes in the residual body.**   To quantify the number of parasites with microneme and rhoptry proteins in the residual body, TgMyoF parasites were transfected with Rop1-NeonFP plasmid, and were grown for 15 hours in either EtOH or IAA before fixation and immunocytochemistry with an anti-AMA1 antibody (S4 Table). The number of vacuoles containing rhoptries or micronemes in the residual body were manually counted. N = 57 vacuoles for AMA-1 quantitation, N = 77 vacuoles for Rop-1 quantitation from two independent experiments. Statistical significance was determined using students t-test.

**Loss of rhoptry positioning at the parasites' apical end.**   To quantify Rop1 localization at the apical or basal ends of the parasite, Rop1-NeonFP was transiently expressed in TgMyo-F-AID parasites treated with EtOH or IAA for 15 hours. Parasites were imaged live as described above. Using the first frame of each movie, the apical half and the basal half of each parasite was outlined manually and the mean fluorescent intensity of Rop1 in the apical and basal ends of the parasites were calculated using Fiji. The apical to basal mean fluorescence intensity was calculated for each parasite. A ratio of 1 indicates that Rop1 is evenly distributed in the apical and basal ends. A ratio >1 indicates more Rop1 is localized in the apical end than the basal end and a ratio <1 indicates more Rop1 in the basal end compared to the apical end. Rop1-NeonFP fluorescence in the residual body was excluded from this analysis. Statistical significance was determined using students t-test.

**Statistics.**   Statistical analyses were performed using GraphPad Prism. Superplots were made as described in [91].

## Supporting information

**S1 Fig. Rab6 and Rop1 dynamics.** Time lapse images of Rab6 (A&B) and Rop1 (C&D) vesicle and compartment dynamics. A and B highlight vesicle (yellow arrowhead) budding from the Rab6 compartment before vesicle movement towards the apical and basal ends of the parasite respectively. Vesicle in A is seen switching direction of movement during imaging. Event highlighted in A, is visible starting at 8.9 seconds from the start of S2 video (control). Event highlighted in B is visible 6.8 seconds from start of the S2 video (control). In C, a vesicle (yellow arrowhead) is observed budding from what is presumably a mature rhoptry before moving towards the parasites basal end. Event highlighted in C is visible at the beginning of S6 Video (control). In D, a large Rop1 positive compartment (yellow arrowhead) is observed moving towards the parasites apical end and fusing with another rhoptry. Event highlighted in D is visible 5 seconds from the start of S6 Video (control). Scale bar = 1μm
(EPS)

**S2 Fig. Creation of TgMyoF-AID parasite line.** (A) Strategy for creating TgMyoF-AID parasite line. Primer binding sites are indicated with arrows. Primer sequences can be found in S3 Table. Predicted PCR product size with F1/R1 and F2/R2 primer pairs are as indicted. (B) Genomic PCR was used to confirm integration of TgMyoF-AID plasmid into the TgMyoF genomic locus. TgMyoF-AID plasmid was derived from TgMyoF-EmGFP plasmid [39]. The yellow box depicts a loxP site inserted into intron 18 in the parental plasmid. This loxP site was utilized for confirming integration into the TgMyoF genomic locus.
(EPS)

**S3 Fig. TgMyoF knockdown results in aberrant apicplast inheritance.** (A&B) Tir1 parental parasites (A) or TgMyoF-AID parasites (B) treated with ethanol or IAA for 15 hours before

fixation and staining with anti-Cpn60 antibody (a marker for the apicoplast; magenta) and anti-IMC1 (a marker for the IMC to delineate the parasite periphery; yellow) and DAPI (cyan). (B) Quantification of the number of apicoplast per parasite in Tir1 or TgMyoF-AID parasites treated with ethanol or auxin for 15 hours. Colored circles are mean values from each independent experiment. Bar graph and error bars are mean ± SEM. Statistical significant was determined using students t-test.
(EPS)

**S4 Fig. IAA treatment does not affect Golgi number or cause Rab6 compartment fragmentation in Tir1 parasites.** (A) Tir1 parasites expressing Grasp55-mCherry (yellow) and EmGFP-P-Rab6 (magenta) were treated with ethanol or IAA for 15 hours before fixation and staining with anti-IMC1 antibody (cyan). (B) Quantification of number of parasites containing either apical/intact or fragmented/distributed Rab6 compartments in Tir1 parental parasites treated with IAA. (C) Quantification of number of Golgi per parasite in Tir1 parasites treated with ethanol or IAA. Results from three independent experiments; experiment 1 = magenta circle, experiment 2 = cyan circle, experiment 3 = grey circle.
(EPS)

**S5 Fig. Rab6 and Syn6 remain colocalized after TgMyoF knockdown.** (A) TgMyoF-AID parasites expressing Syn6-GFP (yellow) and AppleFP-Rab6 (magenta) treated with EtOH (top row) or IAA (middle and bottom rows). *Bottom row*. Inset is magnification of area in the dashed box. Inset scale bar is 1μm. (B) TgMyoF-AID parasites expressing EmGFP-Rab6 (magenta) were treated for 15 hours with ethanol or IAA before fixation and staining with anti-SORTLR antibody (yellow). Rab6 and anti-SORTLR compartments in close proximity are indicated with magenta and yellow arrowheads respectively. After IAA treatment Rab6 and SORTLR compartments often lose their close association (double arrowheads). Scale bar = 5μm.
(EPS)

**S6 Fig. Time course of Rab6 compartment and Golgi fragmentation.** (A) TgMyoF-AID parasites expressing Grasp55-mCherry (yellow) and EmGFP-Rab6 (magenta) were grown overnight in the absence of drug and then treated with IAA for times between 0 minutes and 7 hours. Images taken from 0-, 1.5-, 3- and 6-hour time points are shown. Scale bar = 5μm. (B) Tir1 parental or TgMyoF-AID parasites were grown overnight in the absence of drug then treated for 7 hours with EtOH or IAA. The number of Golgi/parasite was quantified in 50 parasites in each of 3 independent experiments. (C) Tir1 parental or TgMyoF-AID parasites were grown overnight in the absence of drug then treated for 7 hours with EtOH or IAA. The number of parasites with apical/intact Rab6 compartment or redistributed/fragmented was quantified in 50 parasites in each of 3 independent experiments.
(EPS)

**S1 Video.** Live cell imaging of intracellular parasites expressing AppleFP-Rab6 (left panel) and Syntaxin6-GFP (middle panel) and overlay (right panel). Imaging speed was 2 frames/second (fps). Playback speed 15fps. Scale bar = 5μm.
(AVI)

**S2 Video.** RH parasites expressing EmGFP-Rab6 untreated (left), treated with 2μm cytochalasin D for 60 minutes (center), or treated with oryzalin for 18 hours (right). Imaging speed was 10fps. Playback speed 60fps. Scale bar = 5μm.
(AVI)

**S3 Video.** TgMyoF-AID parasites expressing EmGFP-Rab6 were treated with ethanol (left) or IAA (right) for 18 hours before imaging. Imaging speed was 10fps. Playback speed 60fps. Scale bar = 5μm.
(AVI)

**S4 Video.** RH parasites expressing eGFP-HDEL to label the endoplasmic reticulum were treated with DMSO (left), CD (center) or oryzalin (right). Imaging speed was 4fps. Playback speed 30fps. Scale bar = 5μm.
(AVI)

**S5 Video.** TgMyoF-AID parasites expressing eGFP-HDEL were treated with ethanol (left) or IAA (right) for 18 hours before imaging. Imaging speed was 4fps. Playback speed 40fps. Scale bar = 5μm.
(AVI)

**S6 Video.** TgMyoF-AID parasites expressing Rop1-NeonFP were treated with ethanol (left) or IAA (right) for 18 hours before imaging. Imaging speed was 10fps. Playback speed 60fps. Scale bar = 5μm.
(AVI)

**S1 Table. Summary of vesicle trafficking results.** * indicates number of vacuoles instead of number of parasites.
(DOCX)

**S2 Table. Plasmids used in this study.**
(DOCX)

**S3 Table. Primers used in this study.**
(DOCX)

**S4 Table. Antibodies used in this study for immunofluorescence assays and western blots.**
(DOCX)

**S5 Table. ToxoDB ([www.toxodb.org](www.toxodb.org)) accession numbers of genes used in this study.**
(DOCX)

## Acknowledgments

We thank members of the Heaslip Lab, Dr. Ken Campellone and members of the Campellone lab (University of Connecticut) for helpful discussions during the course of these experiments. We thank our colleagues for sharing reagents: Dr. Gary Ward (University of Vermont) for the IMC-1 and AMA-1 antibodies; Dr. Markus Meissner (Ludwig-Maximilian University, Munich) for the DrpB and Syntaxin 6 expression constructs; Dr. Vern Carruthers (University of Michigan) for the TgCPL antibody and the HA-Rab5 and HA-Rab7 expression plasmids; Dr. David Sibley (Washington University, St. Louis) for sharing the Flag-Tir1 parasite line; Dr. Boris Striepen (University of Pennsylvania) for the anti-Cpn60 antibody; Dr. Sabrina Marion (Pasteur Institute, Lille) for the anti-SORTLR antibody.

## Author Contributions

**Conceptualization:** Romain Carmeille, Aoife T. Heaslip.

**Formal analysis:** Romain Carmeille, Porfirio Schiano Lomoriello.

**Funding acquisition:** Aoife T. Heaslip.

**Investigation:** Romain Carmeille, Porfirio Schiano Lomoriello, Parvathi M. Devarakonda, Jacob A. Kellermeier, Aoife T. Heaslip.

**Project administration:** Aoife T. Heaslip.

**Supervision:** Aoife T. Heaslip.

**Visualization:** Romain Carmeille, Porfirio Schiano Lomoriello, Parvathi M. Devarakonda, Aoife T. Heaslip.

**Writing – original draft:** Aoife T. Heaslip.

**Writing – review & editing:** Romain Carmeille, Porfirio Schiano Lomoriello, Parvathi M. Devarakonda, Jacob A. Kellermeier, Aoife T. Heaslip.

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
