## [Decision Letter · Decision Letter 0]

10 Aug 2020

Dear Dr. Heaslip,

Thank you very much for submitting your manuscript "Actin and an unconventional myosin motor, TgMyoF, control the organization and dynamics of the endomembrane network in Toxoplasma gondii." for consideration at PLOS Pathogens. As with all papers reviewed by the journal, your manuscript was reviewed by members of the editorial board and by several independent reviewers. The reviewers appreciated the attention to an important topic. Based on the reviews, we are likely to accept this manuscript for publication, providing that you modify the manuscript according to the review recommendations.

All reviewers commend you on the high standard of imaging and analysis that identifies new roles for MyoF as well as new insights into the complex and distinct endomembrane system of Toxoplasma. The reviewers identified different major concerns (as well as a series of minor concerns), the most urgent being that the Golgi defect could be primary, and downstream endomembrane system defects are secondary due to the Golgi defect. This seems conceivable, but is testable. Additional concerns are that the distinct localization of Rab6 compared to previous reports could be due to the marker used (large tag and overexpression) as well as the request for (key) IAA controls on wild type parasites. These latter two, although important, are less critical for the overall MyoF and endomembrane system message of the manuscript.

Sincerely,

Marc-Jan Gubbels

Reviews Editor and guest AE

PLOS Pathogens

Kami Kim

Section Editor

PLOS Pathogens

Kasturi Haldar

Editor-in-Chief

PLOS Pathogens

orcid.org/0000-0001-5065-158X

Michael Malim

Editor-in-Chief

PLOS Pathogens

orcid.org/0000-0002-7699-2064

All reviewers commend you on the high standard of imaging and analysis that identifies new roles for MyoF as well as new insights into the complex and distinct endomembrane system of Toxoplasma. However, all reviewers identified different concerns, the most urgent being that the Golgi defect could be primary, and downstream endomembrane system defects are secondary due to the Golgi defect. This seems conceivable, but is testable. Additional concerns are that the distinct localization of Rab6 compared to previous reports could be due to the marker used (large tag and overexpression) as well as the request for (key) IAA controls on wild type parasites. These latter two, although important, are less critical for the overall MyoF and endomembrane system message of the manuscript.

Reviewer Comments (if any, and for reference):

Reviewer's Responses to Questions

**Part I - Summary**

Reviewer #1: The study from Carmeille et al. examined the role of the molecular motor TgMyoF in promoting Rab6-positive vesicle dynamics and in regulating endosomal compartment apical positioning. The Authors (Au) showed that the inducible depletion of TgMyoF results in a blockage of Rab6-vesicle motion as well as alterations in the localization and the integrity of Rab5 and Rab7 endosomal compartments. They additionally provided preliminary evidence for a role of TgMyoF in promoting ER tubule dynamics, as well as ROP1-positive vesicle motion, thereby showing the dynamic nature of the rhoptries.

A previous study by Jacot et al., 2013 concluded on the role of TgMyoF in regulating apicoplast inheritance however the present work provided interesting novel insights into the function of this molecular motor on additional intracellular compartment dynamics. The experiments are well conducted with appropriate controls and include quantitative live imaging, which allows to clearly demonstrate the main conclusions proposed by the Au. I am however less convinced about the role of TgMyoF in endosomal compartment apical positioning unless the Au provide further evidence that the defects observed in absence of TgMyoF are not an indirect consequence of Golgi integrity/function alterations (see detailed comments).

Reviewer #2: The goal of this manuscript to determine how the endomembrane system in Toxoplasma is regulated. The authors report that actin and myosin F control ER, Golgi and post golgi compartment dynamics, and rhoptry positioning and trafficking. Overall, the experiments are well conducted and the data are important, novel, and will be of interest to the journal’s general readership. My comments are relatively minor.

Reviewer #3: Toxoplasma gondii parasites contain a highly polarized endomembrane system that comprises both conserved and unique organelles. Past studies have identified numerous post-Golgi compartments in the endomembrane system of the parasite, although the roles, biogenesis and interconnectivity of these post-Golgi compartments remains poorly understood. This manuscript examines the positioning and dynamics of numerous post-Golgi compartments in T. gondii. The data indicate a key role for actin and a myosin termed MyoF in these processes, providing some important and novel insights into the enigmatic endomembrane system of T. gondii. The live cell microscopy approaches, including videos, are of a very high standard, and the microscopy data are quantified wherever feasible. A strength of the paper is that the observational approaches are coupled with genetic perturbations of MyoF, lending an important functional component to the study.

**Part II – Major Issues: Key Experiments Required for Acceptance**

Reviewer #1: Fig 3A:

Rab6 is known to interact with microtubule motors in mammalian cells. Thus, a control with a microtubule destabilizing agent should be included to exclude a role of this cytoskeleton in Rab6 vesicle motion.

Fig 3B: The Au should perform co-localization studies between the fragmented Rab6 compartment and Golgi markers (notably, a marker for the trans-Golgi as the endosomal-like compartments are tightly associated with the trans-Golgi in T. gondii) to verify that the effect they observe is not an indirect consequence of a loss in Golgi structure/integrity upon CD treatment. One may envision that these Golgi fragments may not be able to functionally interact with the actin/myosin machinery and thus remain static.

Although not mandatory (since the depletion of TgMyoF supports a role for the actin cytoskeleton in Rab6 vesicle movement), the Au may consider to use actin KO parasites to confirm the data they obtain using CD treatment.

Fig5: The Au should take advantage of the rapid inducible AID system to improve the characterization of the PGC defects they described upon TgMyoF depletion. In particular, is the dispersion of the studied endosomal compartments (Rab5/Rab7 and Rab6) observed at early time points of TgMyoF depletion (2h, 3h, 4h) and most importantly, is this phenomenon concomitant or independent of (Cis-/Trans-) Golgi fragmentation.

This could be even achieved by live imaging after recording parasites in G1 phase every 30 min during 30-60s for a duration of 4-6 hours with IAA treatment.

Fig 8: If TgMyoF is involved in vesicle transport from and at the Trans-Golgi /ELC, alteration of the Golgi structure/function is expected. Could the Au provide electron microscopy images to further detail the structure of the Golgi (they could also document the other organelles, such as the ER and the rhoptries) in TgMyoF-mAID parasites treated with IAA for 4hours and 15hours?

Reviewer #2: I have two major issues. First, it was surprising that ER tubule dynamics were not examined in the presence of actin and microtubule inhibitors. Even if this was done previously, I think it’s critical to repeat and reproduce these results. Second, if it’s possible, I would suggest supplementing the figures here with additional images acquired using super resolution microscopy.

Reviewer #3: Figure 1B-C. The purpose of the experiments in Figure 1 is to establish the localization of Rab6, and the authors conclude that Rab6 localizes to a different compartment to what was previously reported in the literature. The authors examine Rab6 localization by transiently expressing a fluorescent protein-fused Rab6 from a non-native promoter. Transient expression, the use of a large fusion protein, and a non-native promoter all raise concerns about potential artefacts in the observed localization. The authors should test whether a natively-tagged Rab6 protein exhibits a similar localization. e.g. introduce a (smaller) tag into the native locus of Rab6 using established genome editing approaches.

Figure 5 (and Figures 7-9). The authors should include parental/wild type controls to determine whether the addition of IAA in the absence of TgMyoF depletion influences positioning and appearance of the various post-Golgi compartments that they analyze. I appreciate that it would be considerable work to repeat all the quantificaitons, so perhaps some key controls could be included (e.g. examining the positioning and number of Rab6 compartments in Figure 5, and the positioning/number of Golgi/rhoptries in Figures 8 and 9), with other experiments repeated (and quantified) only if these initial ones reveal concerns about the effects of IAA on processes in the endomembrane system.

**Part III – Minor Issues: Editorial and Data Presentation Modifications**

Reviewer #1: Fig 2: How the directed nature of the Rab6 vesicle motions has been defined? The methodology should be mentionned within the main text. Fig 2D (mentioned in the text) is not part of Fig2?

-Video 1 shows Rab6-positive vesicles emerging from the Golgi/ELC area (anterograde transport). However, video 2 also shows Rab6(+) vesicle fusion at the Golgi (may be retrograde transport). In mammalian cells, Rab6 is involved in retrograde transport between the Golgi and the ER but also in targeting cargos from the Golgi to the plasma membrane. What is the opinion of the Au on the possible functions of Rab6 in T. gondii based on their observations. This should be more clearly discussed in the discussion chapter of the manuscript.

-Video 2 nicely shows bi-directional trajectories of Rab6-positive vesicles along the parasite cortex and numerous vesicles at the basal pole of the parasite. May this reflect an activity for Rab6 in recycling? This activity may regulate rhoptry and microneme compartment homeostasis?

Fig 3A:

-The Au should add a video (as snapshot images) showing the tracking (trajectory shown) of the Rab6-positive vesicles and the corresponding kymographs. The images in Figure 3A present a dynamic Golgi-like structure (rather than single moving Rab6(+) vesicles) in which it is difficult to assess what has been tracked.

Fig 5B: The number of Rab6-positive vesicles increases in TgMyoF depleted parasites whereas it decreases in presence of cytochalasin D. How do the Au reconciliate this two findings?

Fig 6: How the Au explain that the velocity of Rab6-positive vesicles is not affected but only the frequency of directed runs? Which mechanisms regulate their motion in absence of TgMyoF? For instance, a transition from directed to diffusive motion should be correlated with changes in vesicle velocity.

Fig9: The Au wrote in the discussion:

“Our data suggests that TgMyoF is required for movement of immature rhoptries to the apical tip but is not required for TgARO1-dependent anchoring once the organelles have reached their destination.”

Are ROP1 dynamic vesicles co-localizing with any of the endosomal markers used in the study and known to be co-localized with pro-rhoptry proteins? The movie (video 5) shows clear ROP1-positive vesicle transport from the Golgi/ELC area to the apical end of the parasite, likely the mature anchored rhoptries. Some vesicles seem also to bud out the rhoptries? Supporting a role for this transport, in TgMyoF parasites, two almost static ROP1-positive spots (one near the nucleus and one more apical) are observed. The Au could better describe these observations in the “results” chapter page 9.

Discussion:

The Au conclude: “This study demonstrates that TgMyoF controls the dynamics, positioning, and movement of a wide array of organelles in the endomembrane pathway in T. gondii. Future studies will be important to elucidate the mechanism by which this single molecular motor controls the movement of such a wide array of membranous cargos.”

The Au should modulate their conclusions as they demonstrated that TgMyoF controls the movement of Rab6-positive vesicles and ROP1-positive vesicles and that TgMyoF depletion leads to a mis-localization of the Rab5/7 compartment, which could be an indirect consequence of the role of TgMyoF in maintaining Golgi function and integrity during parasite replication.

Reviewer #2: 1. Figure 1B it is not clear whether or not there is overlap between RAB5 and Golgi marker.

2. Syntaxin 6 may be localized to the IMC and not the plasma membrane

3. Figure 2D is referred to in text but is missing.

4. It was unclear to me how organelle rearrangements noted in the manuscript (e.g. line 158) is not due to points of interest moving in and out of the focal plane? Perhaps the authors used a confocal imaging system but this wasn’t clear from the Methods section.

5. I am not sure how but Figure 7 would benefit from quantification.

6. Line 272 should be 8F and 8G

7. Line 279: I think that this is a little strong of a conclusion because it's also possible that centrosome positioning is important for Golgi morphology

Reviewer #3: Lines 100-101. “T. gondii has a single actin gene (TgAct1) that has 83% similarity with chicken skeletal actin (40).” This is an odd comparison to make. Why chicken actin and not that from other animals, or other eukaryotes?

Line 168-169. “the main Rab6(+) compartment lost its apical localization and became fragmented and distributed throughout the parasite cytosol”. How was the “main Rab6+ compartment” identified? In Figure 3B and Video 2, there seems to be a Rab6+ structure apical to the nucleus.

Line 175. “directed runs”. The definition of this wasn’t clear to me. Movement of vesicles in a directed manner? And if so, how was directed movement distinguished from non-directed movement? Likewise, the meaning of “run frequency” in the graph in Figure 3D is unclear. The authors should clarify this. The same point applies to the data in Figure 6 and Figure 9E.

Figure 3D legend (Line 865). “Error bars indicated mean and SEM”. The mean and SEM of the 3 independent experiments or of the combined observations from all the experiments. I presume the former, but the authors should specify this. Similar comment for Figure 5B (Lines 883-884), Figure 6C (Line 897) and Figure 9E-G (Lines 943-944). Strictly speaking for all of these, the error bars show the variation/SEM, not the mean.

Figure S1. The PCR screens that were undertaken, and the associated schematic images, need a clearer description. What are the expected sizes of the various PCR bands? What is the ‘yellow’ box depicted in the bottom schematic? The authors should also include a parental control in these screens (i.e. to determine the sizes/presence of bands in an unmodified locus).

Figure S2B. The authors should include a parental control to determine the effects of IAA on apicoplast inheritance. The authors could also consider including other measures to ascertain whether mAID-mediated MyoF knockdown phenocopies the effects of MyoF loss reported in previous studies (e.g. does knockdown of MyoF-mAID cause a defect in parasite proliferation?)

Figure 8H. How long were parasites incubated in IAA in these experiments? Should specify this in the figure legend or main text.

Lines 346-348. “We have identified Rab6 as a new marker of the syntaxin 6 compartment, a protein that plays a role in retrograde trafficking from the Rab5a/Rab7 compartments to the Golgi”. I think the authors mean that syntaxin 6 plays a role in retrograde trafficking, whereas the text suggests Rab6 does (or that the “syntaxin 6 compartment” is a protein, which doesn’t make sense). This needs to be re-written for clarity.

Lines 354-355. “This data is consistent with a recent report demonstrating that Rab11a is found on the surface of dense granules”. I don’t understand the meaning here. Why would Rab6 not co-localizing with dense granules be consistent with Rab11a localizing with dense granules. The authors should be clearer here.

Lines 394-395. “where there is increased accumulation of rhoptries throughout the parasite cytosol”. I’m not sure it is right to call the ROP1-GFP-labelled structures observed in the MyoF knockdown parasites rhoptries. The data that the authors present in Figure 9 seem to show that the ROP1-GFP protein localizes in smaller vesicles rather than intact, club-shaped rhoptries. As the authors suggest at the end of the paragraph, these might represent immature rhoptries or vesicles containing rhoptry material. Better to reword to “…where this is increase accumulation of the rhoptry protein marker to vesicles throughout the parasite cytosol” or similar.

Minor comments on the text and grammar

Throughout: Golgi (uppercase “G”) vs golgi (lowercase “g”). Should be consistent.

Line 79. “Rab6, is thought to localizes to…”- “thought to localize to”

Line 107-108. “including apicoplast inheritance (a non-photosynthetic plastid organelle)”. Apicoplast inheritance is not an organelle … Perhaps: “including in the inheritance of the apicoplast (a non-photosynthetic plastid organelle)”

Line 185. “referred to subsequently as TgMyoF-AID”. Some of the figures refer to this as “TgMyoF-mAID” (Figure 4 and 6)

Line 191. “Fig. S1B” - should refer to Fig. S2B.

Line 217. “compartment’s” not “compartments”. No full stop before “(Fig. 6A …)”.

Line 271. “twp” – “two”

Line 296. “parasite’s” – no apostrophe

Line 299. “Video 6”. This wasn’t present in my version of the manuscript. Do the authors mean Video 5, which also appears to show moving ROP1 vesicles?

Line 334. “… could be … may be …” - vague. Perhaps just “…is that the Golgi may be comprised of …”

Line 354. “This data …” – “These data …” (plural)

Line 358. “compartments is” – singular or plural? Later in this sentence - “… rearrangement, we observed …” – does the “we” need to start a new sentence?

Line 364. “along BAR domain…” – “along with”?

Line 433. “as indicated in figure 3.” Figure 4, no?

PLOS authors have the option to publish the peer review history of their article (what does this mean?). If published, this will include your full peer review and any attached files.

Reviewer #1: No

Reviewer #2: No

Reviewer #3: No
---

## [Editor Report · Decision Letter 1]

7 Jan 2021

Dear Dr. Heaslip,

We are pleased to inform you that your manuscript 'Actin and an unconventional myosin motor, TgMyoF, control the organization and dynamics of the endomembrane network in Toxoplasma gondii.' has been provisionally accepted for publication in PLOS Pathogens.

Best regards,

Marc-Jan Gubbels

Reviews Editor

PLOS Pathogens

Kami Kim

Section Editor

PLOS Pathogens

Kasturi Haldar

Editor-in-Chief

PLOS Pathogens

orcid.org/0000-0001-5065-158X

Michael Malim

Editor-in-Chief

PLOS Pathogens

orcid.org/0000-0002-7699-2064

The authors have carefully addressed the key concerns and many others.
---

## [Editor Report · Acceptance letter]

27 Jan 2021

Dear Dr. Heaslip,

We are delighted to inform you that your manuscript, "Actin and an unconventional myosin motor, TgMyoF, control the organization and dynamics of the endomembrane network in Toxoplasma gondii.," has been formally accepted for publication in PLOS Pathogens.

Best regards,

Kasturi Haldar

Editor-in-Chief

PLOS Pathogens

orcid.org/0000-0001-5065-158X

Michael Malim

Editor-in-Chief

PLOS Pathogens

orcid.org/0000-0002-7699-2064